# Genome-Wide Analysis of Stress-Responsive Genes and Alternative Splice Variants in *Arabidopsis* Roots under Osmotic Stresses

**DOI:** 10.3390/ijms241914580

**Published:** 2023-09-26

**Authors:** Hye-Yeon Seok, Sun-Young Lee, Swarnali Sarker, Md Bayzid, Yong-Hwan Moon

**Affiliations:** 1Korea Nanobiotechnology Center, Pusan National University, Busan 46241, Republic of Korea; seokhyeon@pusan.ac.kr (H.-Y.S.); symoonlee@pusan.ac.kr (S.-Y.L.); 2Department of Integrated Biological Science, Pusan National University, Busan 46241, Republic of Korea; swarnasarker242@gmail.com (S.S.); md.bayzid18fvm@gmail.com (M.B.); 3Department of Molecular Biology, Pusan National University, Busan 46241, Republic of Korea; 4Institute of Systems Biology, Pusan National University, Busan 46241, Republic of Korea

**Keywords:** alternative splicing, *Arabidopsis*, drought stress, mRNA-Seq, osmotic stress, root, salt stress, transcription factor

## Abstract

Plant roots show distinct gene-expression profiles from those of shoots under abiotic stress conditions. In this study, we performed mRNA sequencing (mRNA-Seq) to analyze the transcriptional profiling of *Arabidopsis* roots under osmotic stress conditions—high salinity (NaCl) and drought (mannitol). The roots demonstrated significantly distinct gene-expression changes from those of the aerial parts under both the NaCl and the mannitol treatment. We identified 68 closely connected transcription-factor genes involved in osmotic stress-signal transduction in roots. Well-known abscisic acid (ABA)-dependent and/or ABA-independent osmotic stress-responsive genes were not considerably upregulated in the roots compared to those in the aerial parts, indicating that the osmotic stress response in the roots may be regulated by other uncharacterized stress pathways. Moreover, we identified 26 osmotic-stress-responsive genes with distinct expressions of alternative splice variants in the roots. The quantitative reverse-transcription polymerase chain reaction further confirmed that alternative splice variants, such as those for *ANNAT4*, *MAGL6*, *TRM19*, and *CAD9*, were differentially expressed in the roots, suggesting that alternative splicing is an important regulatory mechanism in the osmotic stress response in roots. Altogether, our results suggest that tightly connected transcription-factor families, as well as alternative splicing and the resulting splice variants, are involved in the osmotic stress response in roots.

## 1. Introduction

Under osmotic stress, plant roots activate a complex set of physiological and molecular mechanisms that constitute the osmotic stress response [1,2,3]. Owing to their distinct functions and anatomical structures, shoots and roots respond differently to osmotic stress. The primary response of shoots to osmotic stress is the regulation of stomatal conductance [4,5,6,7]. Plants close their stomata to reduce water loss through transpiration in response to osmotic stress, maintaining the cellular hydration state. Conversely, the primary response of roots to osmotic stress is the alteration of water uptake and transport. Under high-solute concentrations, such as in saline soils or during droughts, the water uptake in roots is downregulated, resulting in cellular dehydration. To counteract this dehydration, plants upregulate water uptake and maintain cellular turgor via various mechanisms, such as the accumulation of compatible solutes [5,7,8,9].

When plants are subjected to high salt or drought stress, osmotic stress signals are mainly transmitted through the abscisic acid (ABA)-dependent and -independent pathways [1,2,10]. In the ABA-dependent pathway, osmotic stress increases cellular ABA levels, which induce the expression of osmotic stress-responsive genes, such as *RAB18*, *KIN1*, and *RD29B*, a dehydrin-family protein, an anti-freeze protein, and a CAP160 protein, respectively. In contrast, the osmotic-stress-responsive genes in the ABA-independent pathway are upregulated by the action of the transcription-factor dehydration-responsive element-binding factor 1B/C-repeat-binding factor 1 (DREB1B/CBF1) on C-repeat/dehydration-responsive elements (CRT/DREs) in their promoters. Both signaling pathways activate genes that help maintain cellular homeostasis during osmotic stress [1,2,10].

Water stress and high salinity elicit many similar responses in plants. Several studies on maize (*Zea mays*) and tomatoes (*Solanum lycopersicum*) have shown that ABA can promote the growth of roots and shoots by limiting ethylene biosynthesis, which is activated under water stress [11,12,13]. However, the effects of salinity stress on lateral roots are less straightforward. A previous study showed that mild ionic stress stimulates both the initiation and the emergence of lateral roots and that lateral root emergence in loss-of-function *sos1*, *2*, and *3* mutants is inhibited in response to ionic stress, but not osmotic stress [14]. Although previous studies investigated osmotic stress responses in roots, limited genome-wide studies, restricted to microarray analyses, have been performed.

High-throughput technologies, such as microarray and RNA sequencing (RNA-Seq), have been used to identify osmotic-stress-responsive genes and the associated pathways under high-salinity and drought conditions in the roots of *Arabidopsis* (*Arabidopsis thaliana*), rice (*Oryza sativa*), maize, and alfalfa (*Medicago sativa*) [15,16,17,18,19,20]. However, most previous studies, particularly on *Arabidopsis*, only reported microarray analyses [15,19,20]. The use of RNA-Seq, a high-throughput next-generation sequencing (NGS) technology, has allowed the rapid analysis of large genomic datasets and the quantification of transcriptomes and splice variants. These RNA-Seq analyses have been used to determine global gene-expression patterns in samples at different developmental stages in response to various stimuli and genotypes [21].

The use of pre-mRNA splicing is a crucial step in eukaryotic gene expression. Alternative splicing occurs when splice sites are differentially recognized, resulting in the generation of more than one transcript and, potentially, multiple proteins from the same pre-mRNA. The selection of splice sites under particular cellular conditions is determined by the interaction of globally designated splicing factors, which are proteins that guide spliceosomal components and, therefore, the spliceosome to their respective splice sites [22,23]. Abiotic stresses, such as heat, cold, salt, and drought, markedly alter alternative splicing patterns in plants, and these splicing events induce changes in gene expression for adaptive responses to adverse environments. Splice variants respond distinctly in several respects, such as expression in different tissues or degradation via nonsense-mediated decay [24,25,26]. Alternative splicing mechanisms and the functions of alternative splice variants in roots under osmotic stress conditions have not been well studied.

The existence of distinct osmotic stress responses in shoots and roots suggests that genes responsive to osmotic stresses are explicitly expressed in roots. Therefore, we anticipated that a genome-wide analysis would contribute to identifying novel osmotic-stress-responsive genes and osmotic-stress-responsive mechanisms in roots. In this study, we aimed to determine whether root responses to osmotic stress are mediated by the established osmotic-stress-responsive genes of the ABA-dependent and –independent pathways, or whether other uncharacterized signaling pathways are involved. Additionally, we analyzed whether alternative splicing patterns and the expression of splicing factors differ between roots and shoots. We performed transcriptional profiling of *Arabidopsis* roots under osmotic stress conditions using mRNA-Seq to assess changes in gene expression and to elucidate the molecular mechanisms underlying the osmotic stress response in roots.

## 2. Results

### 2.1. Comparison of Osmotic-Stress-Responsive Genes in Whole Seedlings and Roots

To analyze the regulatory mechanisms of salt- and drought-stress responses in roots, we performed mRNA-Seq analyses (Appendix A). The genes with very low abundance were removed from the analysis, leaving 37,980 genes for further analysis (Appendix A). The NaCl and mannitol treatments replicated the salt and drought stresses, respectively [27,28]. The genes with ≥2-fold or ≤1/2-fold differences in expression with a false discovery rate (FDR) < 0.05 compared to that of the control were considered up- or downregulated, respectively, in response to the NaCl and mannitol treatments. In the roots, 642 and 339 genes were up- and downregulated, respectively, in response to the NaCl treatment, and 605 and 321 genes were up- and downregulated, respectively, in response to the mannitol treatment (Table 1). In contrast, in the whole seedlings, 1202 and 565 genes were up- and downregulated, respectively, under the NaCl treatment, and 1165 and 510 genes were up- and downregulated, respectively, under the mannitol treatment (Table 1). The distribution of these differentially expressed genes (DEGs) was visualized using volcano plots (Appendix A).

To identify altered biological and molecular processes, gene ontology (GO) terms in three categories, namely, biological process (BP), molecular function (MF), and cellular component (CC), were analyzed. Therefore, we performed a GO enrichment analysis using the upregulated genes to determine and compare their functional significance under the two stress conditions tested. Under the NaCl treatment, the upregulated genes in the roots were enriched in the following terms: response to water deprivation, response to abscisic acid, response to wounding, response to salt stress, defense response to fungus, response to oxidative stress, response to jasmonic acid, response to light stimulus, response to cold, and defense response to other organisms in BP; integral component of the membrane, plasma membrane, extracellular region, cytosol, plasmodesma, endoplasmic reticulum (ER), plant-type vacuole, and plant-type cell wall in CC; and transcription-factor activity, transcription regulatory region sequence-specific DNA binding, oxidoreductase activity, sequence-specific DNA binding, heme binding, transmembrane-transport activity, iron-ion binding, UDP-glycosyltransferase activity, and transferase activity in MF (Appendix A). Under the mannitol treatment, the genes upregulated in the roots were enriched in the following terms: response to water deprivation, response to abscisic acid, response to wounding, response to salt stress, defense response to fungus, defense response to bacterium, response to oxidative stress, response to light stimulus, response to cold, and response to osmotic stress in BP; integral component of membrane, plasma membrane, extracellular region, cytosol, plasmodesma, plant-type vacuole, and plant-type cell wall in CC; and protein binding, metal-ion binding, transcription-factor activity, transcription regulatory region sequence-specific DNA binding, oxidoreductase activity, sequence-specific DNA binding, transmembrane-transport activity, protein-heterodimerization activity, ligase activity, and pyridoxal phosphate binding in MF (Appendix A). Under both the NaCl and the mannitol treatment, the upregulated genes in the whole seedlings were enriched in response to the following terms: water deprivation, response to abscisic acid, response to wounding, defense response to bacterium, and response to salt stress in BP; cytoplasm, membrane, integral component of membrane, ER, and vacuole in CC; and protein binding, transcription-factor activity, oxidoreductase activity, transferase activity, and transmembrane-transporter activity in MF (Appendix A). These data indicate that osmotic stress responses in roots may involve, among other pathways, hormone signaling and antioxidant regulation.

### 2.2. Identification of Osmotic-Stress-Responsive Genes in Roots

We compared the DEGs in the roots and the whole seedlings. Although 390 genes were upregulated in both the roots and whole seedlings under the NaCl treatment (i.e., commonly upregulated genes), 252 and 812 genes were only upregulated in the roots and whole seedlings, respectively (Appendix A). Similarly, 334 genes were upregulated in both the roots and the whole seedlings under the mannitol treatment (i.e., commonly upregulated genes), while 271 and 831 genes were only upregulated in the roots and whole seedlings, respectively (Appendix A). In addition, 85 genes were downregulated under the NaCl treatment in both the roots and the whole seedlings (i.e., commonly downregulated genes), whereas 254 and 480 genes were only downregulated in the roots and whole seedlings, respectively (Appendix A). Moreover, 67 genes were downregulated in both the roots and the whole seedlings under the mannitol treatment (i.e., commonly downregulated genes), whereas 254 and 443 genes were only downregulated in the roots and whole seedlings, respectively (Appendix A). We performed hierarchical clustering to identify the association between the DEGs in different conditions. The up- and downregulated genes were classified into individual hierarchies (Figure 1a). These results were consistent with those from previous studies [19,20]. Moreover, the DEGs in the roots and whole seedlings under the NaCl and mannitol treatments were also classified into individual hierarchies (Figure 1a), indicating that the osmotic stress responses in the roots were distinct from those in the aerial parts.

We analyzed the DEGs in the roots under salt- and drought-stress conditions to elucidate the osmotic stress responses in the roots. We found that 361 genes were upregulated both in the roots treated with NaCl and in those treated with mannitol (i.e., commonly upregulated genes), whereas 281 and 244 genes were only upregulated in the roots treated with NaCl or mannitol, respectively (Figure 1b). In contrast, 193 genes were downregulated both in the roots treated with NaCl and in those treated with mannitol (i.e., commonly downregulated genes), whereas 146 and 128 genes were only downregulated in the roots treated with NaCl or mannitol, respectively (Figure 1c). To identify functional pathways in the salt- and drought-stress responses in the roots, we subjected the genes upregulated in the roots to a Kyoto Encyclopedia of Genes and Genomes (KEGG) pathway-enrichment analysis. We revealed that the genes upregulated in the roots treated with NaCl and mannitol were mainly involved in the metabolic pathway (Figure 1d–f). In addition, several genes upregulated under the mannitol treatment were involved in the biosynthesis of secondary metabolites (Figure 1f). These results indicate that although the salt- and drought-stress responses share some pathways, distinct stress-response pathways are also involved in the responses to different stress conditions.

We analyzed the expressions of well-known ABA-dependent and/or ABA-independent osmotic stress-responsive genes in the roots using mRNA-Seq [10,29,30,31,32,33,34,35,36]. Among the 11 ABA-dependent-pathway-stress-responsive genes, the expression of *RAB18* was markedly increased in the roots compared to that in the whole seedlings. In contrast, the expressions of the 10 remaining genes were only marginally increased in the roots (Figure 2 and Appendix A). Furthermore, the expressions of three ABA-independent osmotic-stress-responsive genes and five ABA-dependent and -independent genes increased less in the roots than in the whole seedlings (Figure 2 and Appendix A), indicating that these well-known osmotic-stress-responsive genes are not key players in osmotic stress responses in roots. Similarly, the expression analysis of the well-known ABA-dependent and/or ABA-independent osmotic-stress-responsive genes in different plant tissues using Genevestigator, a gene-expression resource, showed that their expressions in the root tissues were similar to or lower than those in the shoot tissues (Appendix A).

### 2.3. Identification of Osmotic-Stress-Responsive Transcription-Factor Genes in Roots

To characterize the salt- and drought-stress signal transduction in the roots, we analyzed the genes upregulated under the NaCl and mannitol treatments in the roots and identified 68 upregulated genes that were enriched in transcription-factor-related GO terms, such as “DNA-templated transcription” (GO:0006351) and “regulation of DNA-templated transcription” (GO:0006355) and classified them into 22 transcription-factor families, including MYB, bZIP, NAC, AP2/ERF, WRKY, bHLH, IAA, B-box zinc finger, and HD-Zip, among others (Table 2). The MYB family was the most heavily represented family (fourteen genes), followed by the NAC family (seven genes) and the AP2/ERF, WRKY, and bZIP families (six genes each) (Table 2). These transcription factors may be involved in salt- and drought-stress signal transduction in roots.

To understand the relationships between the osmotic-stress-responsive transcription factors in the roots, a protein network including all 68 transcription factors was mapped using Cytoscape STRING Apps. This analysis showed that 45 transcription factors were tightly connected and interacted with each other. Notably, NAC- and MYB-family proteins, such as NAC032, NAC083, NAC102, MYB3, and MYB108, interacted with multiple transcription factors (Figure 3).

To validate the mRNA-Seq results using quantitative reverse-transcription polymerase chain reaction (qRT-PCR), we selected five transcription-factor genes, namely AT1G62975, *AITR5*, *WRKY29*, *BOA*, and *MYB3*, which belong to the bHLH, DRG, WRKY, GARP, and MYB families, respectively. Accordingly, these genes showed higher expression levels in the roots compared with the other selected transcription-factor genes (Appendix A). We further analyzed the expressions of these five genes in the whole seedlings and roots under the NaCl treatment, using *RAB18* as a positive control. The *RAB18* expression was significantly increased under the NaCl treatment in the whole seedlings and roots (Figure 4a and Appendix A). Similarly, the expressions of the five selected genes increased significantly under salt-stress conditions in the whole seedlings and roots (Figure 4 and Appendix A). These results were consistent with those of our mRNA-Seq analysis (Appendix A). In addition, the expressions of the five selected genes were significantly increased in the whole seedlings and roots under the mannitol treatment (Figure 5 and Appendix A), confirming the validity of the mRNA-Seq analysis (Appendix A). To compare the expressions of the five selected genes in the roots with those in other tissues, we performed qRT-PCR using whole seedlings, shoots, and roots under the NaCl and mannitol treatments. The expressions of AT1G62975, *WRKY29*, and *BOA* were higher in the roots than in the whole seedlings and shoots (Figure 4 and Figure 5), suggesting that these three genes may have important functions in salt- and drought-stress responses in roots.

Among the 68 transcription factor genes, 23 exhibited higher expression levels in the roots than in the whole seedlings under the NaCl and mannitol treatments (Appendix A). To identify signal transduction and the protein–protein interaction networks in these 23 transcription factors, we analyzed protein–DNA- and protein–protein-interaction networks using the Bio-Analytic Resource for Plant Biology (BAR). The results indicated that MP may interact with 16,726 genes, showing the highest number of protein–DNA interactions among the 23 transcription factors analyzed (Appendix A), and that MYB108, MYB12, MYB71, WRKY23, MYB3, TEM1, COL9, MYB4, and FAR1 can interact with 1910; 2, 3, 80, 3016, 43, 3, 191, and 748 genes, respectively (Appendix A). In addition, MP had 36 predicted interaction-partner proteins (Appendix A). Furthermore, IAA18, IAA2, IAA13, MYB12, FAR1, bZIP9, and BOA showed 34, 47, 41, 16, 13, 24, and 19 interaction partners, respectively (Appendix A). These results showed that isolated transcription-factor genes may have an important function in the osmotic stress response in roots through signal transduction by DNA and protein–protein interactions. No protein–DNA or protein–protein interactions were predicted for AITR5, WRKY29, GATA2, LNK4, or GATA12 (Appendix A).

### 2.4. Identification of Osmotic-Stress-Responsive Alternative Splice Variants in Roots

Abiotic stresses, including osmotic stress, can alter splicing patterns in plants, and these splicing events induce changes in gene expression that are crucial for adaptive responses to adverse environments. To elucidate alternative splicing patterns in the osmotic-stress-responsive genes in the roots, we analyzed the expressions of alternative splice variants using mRNA-Seq. Several alternative splice variants of osmotic-stress-responsive genes exhibited different expression patterns in the roots under the NaCl and mannitol treatments (Figure 6 and Appendix A). For example, four alternative splice variants of *ZIFL1* were observed under both the NaCl and the mannitol treatment in the roots, with *ZIFL1.2* displaying the highest expression (Figure 6 and Appendix A). Among the two alternative splice variants of *ANNAT4*, the expression of *ANNAT4.1* was more than 6-fold higher than that of *ANNAT4.2* (Figure 5 and Appendix A). Similarly, of the two alternative splice variants of AT1G71000, the expression of *AT1G71000.1* was more than 3-fold higher than that of *AT1G71000.2*, and of the two alternative splice variants of *TRM19*, the expression of *TRM19.1* was more than 3-fold higher than that of *TRM19.2* under both the NaCl and the mannitol treatment in the roots (Figure 6 and Appendix A). These findings collectively suggest that alternative splice variants respond distinctly to salt and drought stress in roots.

To validate the expressions of the alternative splice variants in the roots under the NaCl and mannitol treatments, we selected four genes—*ANNAT4*, *MAGL6*, *TRM19*, and *CAD9*—that showed significant differences in expression between alternative splice variants in the roots under the NaCl and mannitol treatments (Figure 6 and Appendix A), and we performed qRT-PCR for the alternative splice variants. The *ANNAT4.1*, an alternative splice variant of *ANNAT4*, was highly upregulated in the roots under the NaCl and mannitol treatments, while no significant differences were observed for the *ANNAT4.2* (Figure 7a and Figure 8a). Similarly, the expressions of *MAGL6.1*, *TRM19.1*, and *CAD9.1* increased under the NaCl and mannitol treatments, whereas those of *MAGL6.2*, *TRM19.2*, and *CAD9.2* did not change (Figure 7b–d and Figure 8b–d). The expressions of *ANNAT4.1*, *MAGL6.1*, *TRM19.1*, and *CAD9.1* were significantly higher than those of their corresponding alternative splice variants, *ANNAT4.2*, *MAGL6.2*, *TRM19.2*, and *CAD9.2*, respectively (Figure 7 and Figure 8), suggesting that highly expressed alternative splice variants may play an important role in salt- and drought-stress responses in *Arabidopsis* and not in other alternative splice variants.

## 3. Discussion

In this study, we identified stress-responsive genes and alternative splice variants involved in the osmotic stress response in *Arabidopsis* roots using an mRNA-Seq analysis. We identified 642 and 605 genes that were upregulated in the roots under NaCl and mannitol treatments, respectively (Table 1). In comparison, 339 and 321 genes were downregulated in the roots under the NaCl and mannitol treatments, respectively. Among these DEGs, 361 and 193 genes were up- and downregulated in the roots under both the NaCl and the mannitol treatment, respectively (Figure 1b,c), indicating that the stress-response pathways may overlap in roots. Among these overlapping DEGs, we identified 22 transcription-factor families involved in osmotic-stress-signal transduction in roots (Table 2 and Appendix A). In addition, we identified alternative splice variants that respond distinctly to NaCl and mannitol treatments in roots, such as those for *ANNAT4*, *MAGL6*, *TRM19*, and *CAD9* (Figure 6, Figure 7 and Figure 8 and Appendix A).

Although roots and shoots share common responses to osmotic stress, they possess unique adaptations. Specifically, in response to osmotic stress, roots primarily respond by altering water uptake and transport. In contrast, shoots respond by regulating stomatal conductance to conserve water [1,2,3]. To understand the osmotic stress response in roots, we performed an mRNA-Seq analysis on *Arabidopsis* roots. The expressions of well-known ABA-dependent and/or ABA-independent osmotic-stress-responsive genes, including *RD29B*, *RD22*, *RD20*, *RD29A*, *COR47*/*RD17*, *DREB2A*, and *DREB2B*, were either lower or marginally different in the roots compared to those in whole seedlings (Figure 2 and Appendix A). This strongly implies that the response to osmotic stress in roots may be mediated by pathways or mechanisms that have not yet been characterized and are distinct from well-known osmotic-stress-responsive pathways. Subsequently, using the mRNA-Seq analysis, we identified 68 transcription-factor genes that were highly expressed in the roots under the NaCl and mannitol treatments (Table 2). These 68 genes were members of 22 transcription-factor families, such as including MYB, bZIP, NAC, AP2/ERF, and WRKY, (Table 2). The *MYB71* gene, which belongs to the MYB family, is reportedly hypersensitive to ABA, suggesting that *MYB71* functions as a positive regulator of the salt-stress response in roots in an ABA-dependent manner [37]. Our BAR analysis suggested that MYB71 binds to the promoters of three genes: *FMO GS-OX1*, *SUR1*, and *GSTF9* (Appendix A). The *SUR1* gene is involved in lateral root development and auxin production [38], indicating that MYB71 may be involved in salt- and drought-stress responses in roots by regulating root-development-related genes, such as *SUR1*. Furthermore, *MYB108* has been shown to be involved in both biotic and abiotic stress responses, such as salinity and drought. Mutants of *myb108* exhibit hypersensitivity to salt and drought stress [39]. In addition, the expressions of cell-wall-related genes were downregulated in *myb108* mutants under combined biotic and abiotic stress conditions [39], implying that *MYB108* is perhaps involved in the osmotic stress response in roots through cell-wall-biosynthesis regulation. In contrast to MYB71, MYB3 has been identified as a transcriptional repressor [40]. Consistently, *myb3* mutants exhibit enhanced root growth and high accumulations of lignin and anthocyanins under salt-stress conditions [40]. The *AITR5* gene is hyposensitive to ABA, and *aitr5* mutants are tolerant to drought stress [41]. Furthermore, AITR5 represses the ABA-repressed expression of receptor genes and the ABA-induced expression of PP2Cs, which function as negative-feedback-regulation loops in ABA signaling [41]. In addition, *AITR5* may function in the osmotic stress response in roots in an ABA-dependent manner. Furthermore, *MYB12* has been identified as being involved in flavonoid biosynthesis, especially in roots. It acts on root-hair elongation by negatively regulating cell vascular proliferation to optimize the cell-proliferation rate during root vascular development [42], suggesting that it may be involved in the osmotic stress response through the regulation of root-hair development. We speculate that the identified transcription-factor genes may lead to new research on the identification of novel mechanisms or signaling pathways associated with the osmotic stress response in roots. Previously, meta-analysis studies using microarrays were performed to understand osmotic stress responses in roots and focused on transcription-factor genes [15,19,20]. In roots, transcription-factor genes are involved in osmotic stress responses in various ways, such as cell-wall modification, osmoprotective synthesis and transport, reactive oxygen species (ROS) scavenging, protein metabolism, and hormone signaling [15,19,20]. Further studies on identified transcription-factor genes provide clues to understanding osmotic stress responses and root regulatory mechanisms.

Although the regulation and functions of alternative splice variants have been studied in plants under several abiotic stress conditions, their role in roots under abiotic stress conditions is yet to be explored. Moreover, previous meta-analysis studies on *Arabidopsis* roots under stress conditions were mostly limited to microarray analyses, which could not be used to analyze alternative splice variants [15,19,20]. Using an mRNA-Seq analysis, we identified 26 genes that showed varying expressions of alternative splice variants or altered splicing events in roots under NaCl and mannitol treatments (Figure 6 and Appendix A). Furthermore, the expression of the alternative splice variants of *ANNAT4*, *MAGL6*, *TRM19*, and *CAD9* in the roots significantly differed under the NaCl and mannitol treatments (Figure 7 and Figure 8). The *ANNAT4* gene is involved in oxidative stress responses through calcium-mediated signal transduction in roots [43,44]. Osmotic stress, especially at high salinity, produces ROS [45], indicating that *ANNAT4* is involved in the osmotic stress response in roots via the regulation of the oxidative stress response. Transcription-factor genes like *MYB3* and *IAA2* have alternative splice variants (Figure 6 and Appendix A). These variants showed differential expression in the roots under the NaCl and mannitol treatments (Figure 6 and Appendix A), implying that the alternative splice variants of transcription-factor genes, such as *MYB3* and *IAA2*, result in different sets of downstream genes and/or differential expression levels of downstream genes in roots.

Splice-site selection is determined by core spliceosomal components, along with additional RNA-binding proteins, such as serine/arginine-rich (SR) proteins and heterogeneous nuclear ribonucleoproteins (hnRNPs), which bind to *cis*-regulatory elements in either introns or exons, thereby activating or repressing splicing and resulting in alternative splicing [22,23]. The RZ-1a, an hnRNP, negatively regulates early development under salt- and drought-stress conditions [46]. In contrast, GRP7, another hnRNP, exerts a positive effect on stress tolerance at low temperatures and a negative effect under salt- or drought-stress conditions [47]. Furthermore, a T-DNA insertional mutant of *SR45*, an SR-protein gene, enhances sensitivity to salt stress and changes the expressions and splicing patterns of genes involved in the regulation of the salt-stress response [48,49], suggesting that SR45 positively regulates salt tolerance. Although previous studies investigated splicing factors in the osmotic stress response, alternative splicing mechanisms in roots remain unclear. Hence, we analyzed mRNA-Seq data to identify alternative splicing mechanisms in roots under osmotic stress conditions; however, well-known osmotic-stress-responsive splicing factors, such as *RS40*, *RS41*, *GRP7*, *RZ-1a*, *CBP20*, and *CBP80*, did not exhibit any expression differences in the roots under the NaCl and mannitol treatments (Appendix A). Further studies are therefore required to identify alternative splicing mechanisms in roots.

In summary, we identified osmotic-stress-responsive genes in *Arabidopsis* roots using an mRNA-Seq analysis. We observed that many transcription-factor families are involved in the osmotic stress response in roots and are tightly connected to each other. In addition, alternative splicing and alternative splice variants are important in the osmotic stress response in *Arabidopsis* roots. Further studies of the biological and molecular functions of the identified transcription-factor genes and alternative splice variants will provide valuable and insightful information about the stress responses of *Arabidopsis*. Furthermore, these genes can be potential candidates for generating useful crop traits.

## 4. Materials and Methods

### 4.1. Plant Materials and Growth Conditions

*Arabidopsis thaliana* accession Col-0 background was used in this study. Seeds were sterilized and germinated as previously described [50]. The seedlings were grown under short-day (SD) conditions (8 h light:16 h dark photoperiod) at 22 °C.

### 4.2. Plant Stress Treatment

Ten-day-old Col-0 seedlings grown under SD conditions were placed on filter paper soaked in an MS solution including 150 mM NaCl or 300 mM mannitol. After 0, 1, 2, 4, and 8 h, the seedlings were harvested and prepared as whole seedlings or cut into shoots and roots. Seedlings, shoots, and roots harvested at 0 h were used as controls.

### 4.3. RNA Isolation and First-Strand cDNA Synthesis

Total RNA extraction was conducted using an RNAqueous RNA Isolation Kit (Invitrogen, Carlsbad, CA, USA) and a Plant RNA Isolation Aid (Invitrogen, Carlsbad, CA, USA), according to the manufacturer’s instructions. Subsequently, 2 μg of total RNA was reverse-transcribed in a total volume of 25 μL containing 0.5 μg of oligo dT primer, 0.5 mM dNTPs, and 200 units of Moloney murine leukemia virus reverse transcriptase (Promega Corp., Madison, WI, USA).

### 4.4. Quantitative RT-PCR

The qRT-PCR analysis was performed using Power SYBR^TM^ Green PCR Master Mix (Applied Biosystems, Foster City, CA, USA) and a QuantStudio^TM^ 3 real-time PCR system (Applied Biosystems, Foster City, CA, USA), as previously described [50]. The Ct (cycle at the threshold) value was set as constant throughout the study and corresponded to the log-linear range of PCR amplification. The expressions of the target genes, which reflect the relative expressions of the target transcripts, were normalized to that of the endogenous reference gene, *GAPc*. The experiment was performed with at least two biological replicates, with two technical replicates for each biological replicate. Three independent reactions were performed for each technical replicate. The primers used in this study are listed in Appendix A.

### 4.5. Library Preparation and RNA-Seq

The RNA-Seq was performed on samples from 10-day-old whole seedlings and roots. Whole seedlings were treated with 150 mM NaCl for 1, 2, and 4 h or 300 mM mannitol for 1, 2, and 4 h each. Subsequently, the samples were harvested, and either complete (whole seedlings) or fractionated (roots only) samples were used for RNA isolation. After total RNA isolation, 2 μg of each sample was mixed and used for mRNA-Seq library preparation, which was performed by E-biogen (https://www.e-biogen.com, accessed on 13 September 2022), as previously described [51]. High-throughput paired-end 100 bp sequencing was performed using a HiSeq X10 system (Illumina, Inc., San Diego, CA, USA). Two biological replicates of each sample were used for RNA-Seq.

### 4.6. mRNA-Seq Data Analysis

The mRNA-Seq data analysis was performed by E-biogen (https://www.e-biogen.com, accessed on 13 September 2022). Quality control of raw sequencing data was performed using FastQC [52]. The adapter and low-quality reads (<Q20) were removed using FASTX_Trimmer and BBMap [53,54]. Trimmed reads were then mapped to the reference genome (*Arabidopsis* genome sequence TAIR 10) using TopHat [55]. Gene-expression levels were estimated using their FPKM (fragments per kb per million reads) values, as determined by Cufflinks [56]. The FPKM values were normalized based on the quantile normalization method using EdgeR in R [57]. The mapping rates of RNA-Seq were from 90.2% to 98.0%. The numbers of mapped reads ranged from 27.3 to 50.9 million. The alignment rates were from 88.1% to 95.9% (Appendix A). Data mining and graphic visualization were performed using ExDEGA (E-biogen, Inc., Seoul, Republic of Korea). Statistical analysis was performed using the FDR. The complete mRNA-Seq data from this study were submitted to the Gene Expression Omnibus database (http://www.ncbi.nlm.nih.gov/geo, accessed on 5 April 2023) under the accession number GSE229217. Genes with ≥2-fold and ≤1/2-fold differences in expression with FDR < 0.05 compared to that of the control (whole seedlings and roots under no-treatment condition) were considered up- and downregulated, respectively. A no-treatment condition was used as a control. A GO annotation enrichment was performed using DAVID (http://david.abcc.ncifcrf.gov, accessed on 1 September 2023) with the default parameters [58]. A KEGG pathway analysis was conducted using KEGG Mapper (https://www.genome.jp/kegg/tool/map_pathway2.html, accessed on 1 September 2023) [59]. Gene clustering was performed using MeV v.4.9.0 (https://sourceforge.net/projects/mev-tm4, accessed on 3 May 2023) [60]. Protein-network analysis was performed using StringApp in Cytoscape v.3.9.1 (https://apps.cytoscpae.org/apps/stringapp, accessed on 3 May 2023) [61].

### 4.7. Statistical Analysis

Statistical analysis was performed using one-way analysis of variance (ANOVA) with Tukey’s multiple comparison test using IBM SPSS v.23 (IBM Corp., Armonk, NY, USA).

## Figures and Tables

**Figure 1 ijms-24-14580-f001:**
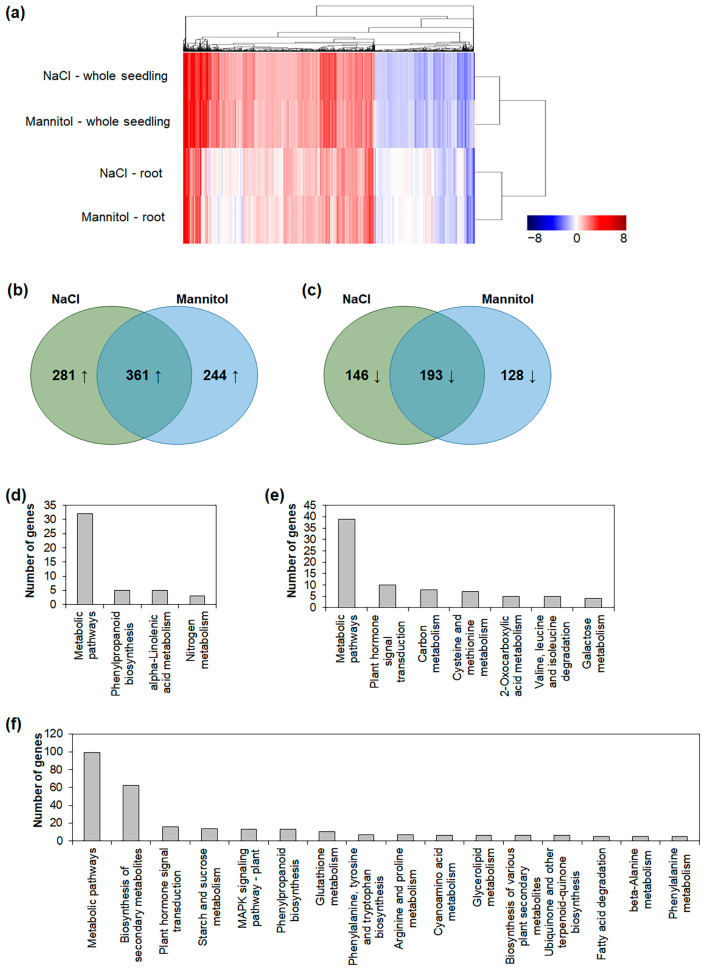
Hierarchical clustering, Venn diagrams and KEGG mapper of DEGs in roots and whole seedlings. (**a**) Hierarchical clustering of upregulated and downregulated genes was performed using MultiExperiment Viewer (MeV). Euclidean distance and average linage clustering were used for hierarchical clustering. (**b**) Venn diagram of upregulated genes in roots under NaCl and mannitol treatments. (**c**) Venn diagram of downregulated genes in roots under NaCl and mannitol treatments. (**d**) KEGG pathway of upregulated genes in roots under both NaCl and mannitol treatments. (**e**) KEGG pathway of upregulated genes in roots under NaCl treatment. (**f**) KEGG pathway of upregulated genes in roots under mannitol treatment. In (**d**–**f**), enriched KEGG pathway shown with *p* < 0.05.

**Figure 2 ijms-24-14580-f002:**
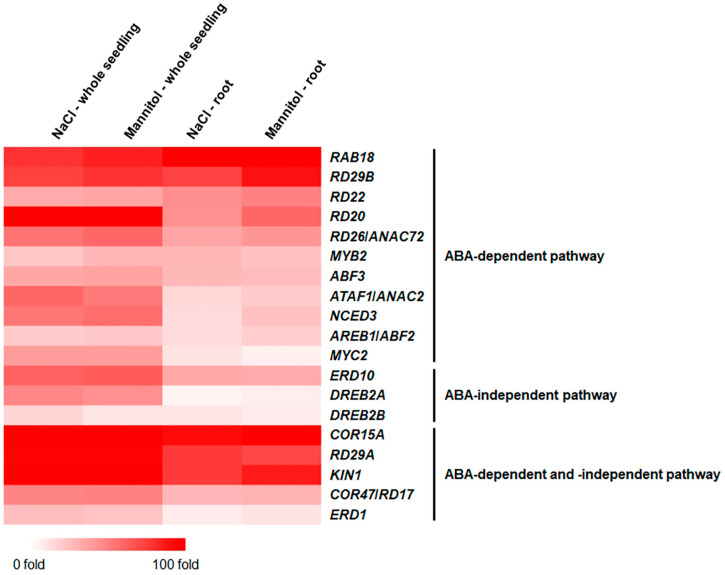
Expressions of the ABA-dependent and -independent osmotic stress-responsive genes in roots. Expressions of ABA-dependent and -independent osmotic stress-responsive genes were visualized using MeV. MeV was performed using two-color array and *Arabidopsis thaliana* organism.

**Figure 3 ijms-24-14580-f003:**
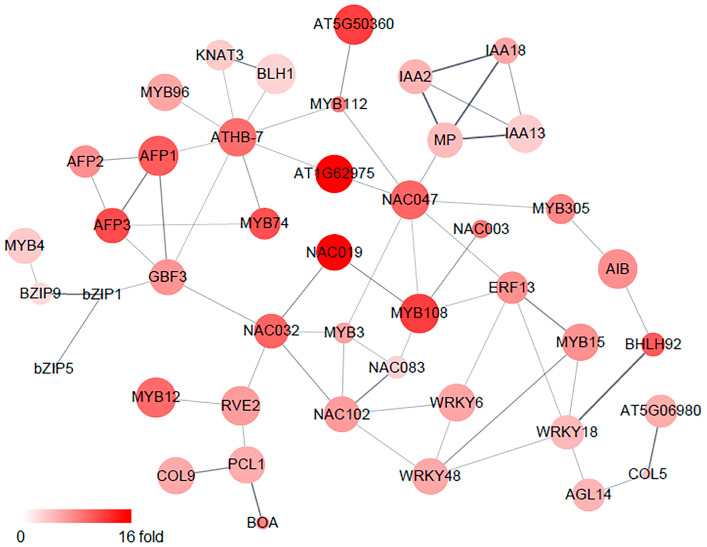
Protein network of the transcription-factor families of upregulated genes in roots. Protein-network interaction of the transcription-factor genes was analyzed using STRING Apps of Cytoscape. STRING analysis was performed using protein queries with gene symbols as identifiers, confidence score of 0.4, and maximum additional interactors of 0. Node color and size were set as fold change and FDR, respectively. Edge thickness represents the confidence in association between two connected nodes and ranges from 0.4 to 1.0, as determined by STRING.

**Figure 4 ijms-24-14580-f004:**
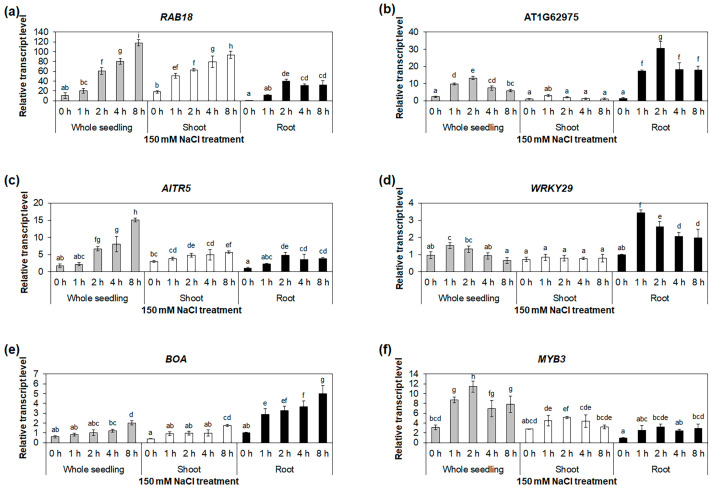
Expression analysis of salt-stress-responsive transcription-factor genes. Quantitative RT-PCR analyses of *RAB18* (**a**), AT1G62975 (**b**), *AITR5* (**c**), *WRKY29* (**d**), *BOA* (**e**), and *MYB3* (**f**) in whole seedlings, shoots, and roots under 150 mM NaCl treatment for 0, 1, 2, 4, and 8 h. The *GAPc* was used as an internal control. Transcript levels at 0 h in roots were set as 1. Three biological replicates were performed, with two technical replicates for each biological replicate. Three independent reactions were performed for each technical replicate. Error bars represent standard deviation (*n* = 6 reactions). Different letters indicate statistically significant differences (*p* < 0.05).

**Figure 5 ijms-24-14580-f005:**
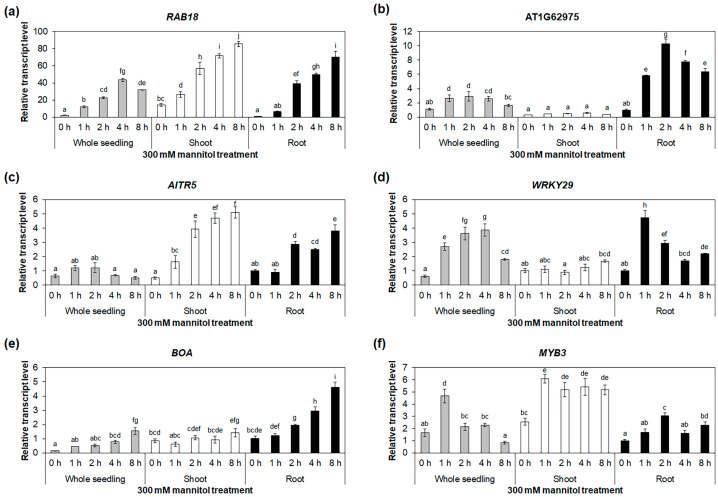
Expression analysis of drought-stress-responsive transcription-factor genes. Quantitative RT-PCR analysis of *RAB18* (**a**), AT1G62975 (**b**), *AITR5* (**c**), *WRKY29* (**d**), *BOA* (**e**), and *MYB3* (**f**) in whole seedlings, shoots, and roots under 300 mM mannitol treatment for 0, 1, 2, 4, and 8 h. The *GAPc* was used as an internal control. Transcript levels at 0 h in roots were set as 1. Two biological replicates were performed with two technical replicates for each biological replicate. Three independent reactions were performed for each technical replicate. Error bars represent standard deviation (*n* = 4 reactions). Different letters indicate statistically significant differences (*p* < 0.05).

**Figure 6 ijms-24-14580-f006:**
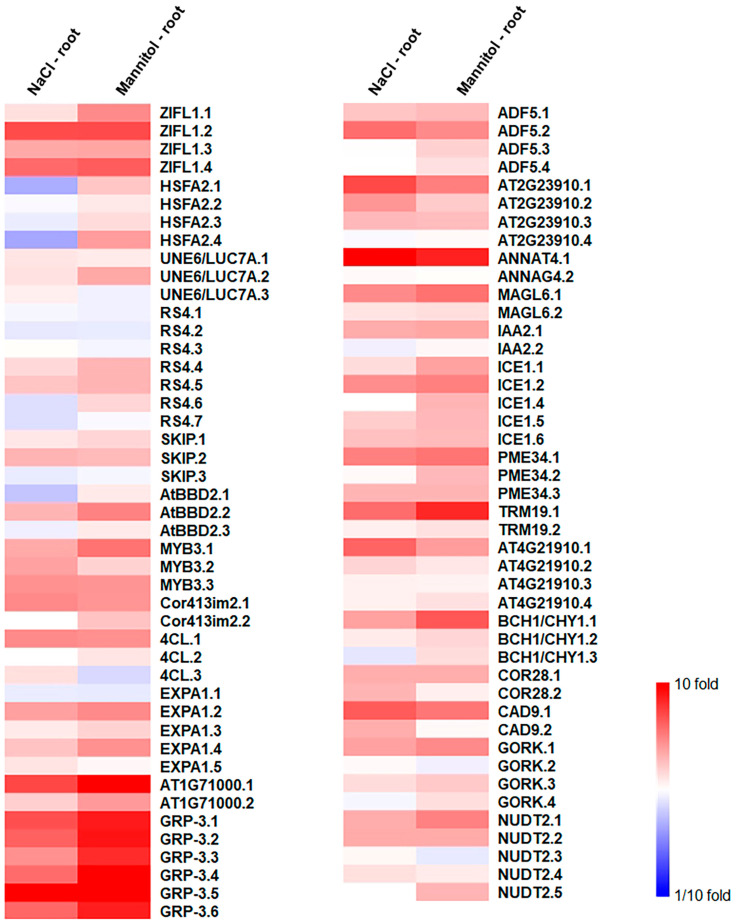
Expressions of upregulated alternative splice variants in roots. Expressions of upregulated alternative splice variants were visualized using MeV. MeV was performed using two-color array and *Arabidopsis thaliana* organism.

**Figure 7 ijms-24-14580-f007:**
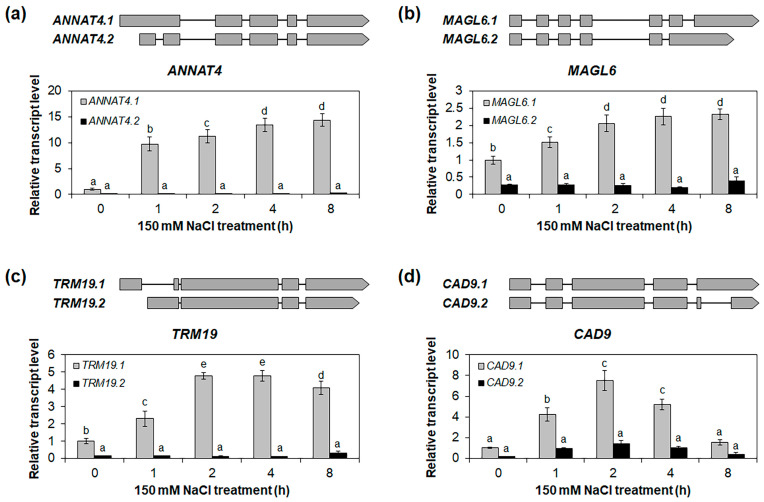
Expression analysis of salt-stress-responsive alternative splice variants in roots. Quantitative RT-PCR analysis of alternative splice variants of *ANNAT4* (**a**), *MAGL6* (**b**), *TRM19* (**c**), and *CAD9* (**d**) in roots under 150 mM NaCl treatment for 0, 1, 2, 4, and 8 h. The *GAPc* was used as an internal control. The relative transcript levels were determined with respect to the transcript level of isoform 1 at 0 h. Isoform 1 of each gene was indicated as the representative isoform on TAIR website (TAIR, https://www.arabidopsis.org, accessed on 22 June 2023). Three biological replicates were performed, with two technical replicates for each biological replicate. Three independent reactions were performed for each technical replicate. Error bars represent standard deviation (*n* = 6 reactions). Different letters indicate statistically significant differences (*p* < 0.05).

**Figure 8 ijms-24-14580-f008:**
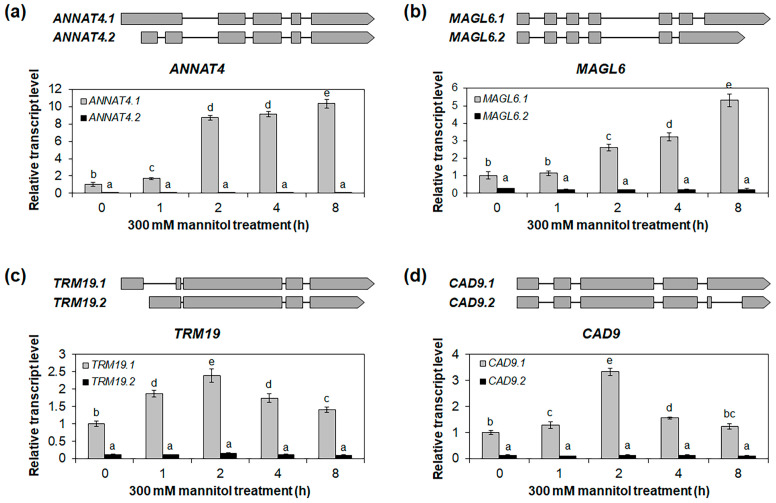
Expression analysis of drought-stress-responsive alternative splice variants in roots. Quantitative RT-PCR analysis of alternative splice variants of *ANNAT4* (**a**), *MAGL6* (**b**), *TRM19* (**c**), and *CAD9* (**d**) in roots under 300 mM mannitol treatment for 0, 1, 2, 4, and 8 h. The *GAPc* was used as an internal control. The relative transcript levels were determined with respect to the transcript level of isoform 1 at 0 h. Isoform 1 of each gene was indicated as the representative isoform on TAIR website (TAIR, https://www.arabidopsis.org, accessed on 22 June 2023). Two biological replicates were performed, with two technical replicates for each biological replicate. Three independent reactions were performed for each technical replicate. Error bars represent standard deviation (*n* = 4 reactions). Different letters indicate statistically significant differences (*p* < 0.05).

**Table 1 ijms-24-14580-t001:** Number of differentially expressed genes (DEGs).

Experiments	Upregulated Gene Number	Downregulated Gene Number
NaCl—whole seedling	1202	565
Mannitol—whole seedling	1165	510
NaCl—root	642	339
Mannitol—root	605	321

**Table 2 ijms-24-14580-t002:** List of transcription-factor families of upregulated genes under osmotic stress conditions in roots.

TF Family	Number of Genes	Genes
MYB	14	*MYB3*, *MYB4*, *MYB12*, *MYB15*, *MYB34*, *MYB41*, *MYB71*, *MYB74*, *MYB96*, *MYB108*, *MYB112*, *MYB122*, *PCL1*, *RVE2*
NAC	7	*NAC003*, *NAC019*, *NAC032*, *NAC047*, *NAC083*, *NAC089*, *NAC102*
AP2/ERF	6	*ABR1*, *ABS2*, *ERF13*, *SMZ*, *TEM1*, AT3G11580
WRKY	6	*WRKY6*, *WRKY18*, *WRKY23*, *WRKY29*, *WRKY31*, *WRKY48*
bZIP	5	*bZIP1*, *bZIP5*, *bZIP7*, *bZIP9*, *GBF3*
bHLH	4	*AIB*, *bHLH92*, *NAI1*, AT1G62975
IAA	4	*IAA2*, *IAA13*, *IAA18*, *MP*
B-box zinc finger	3	*COL5*, *COL9*, *LNK4*
HD-Zip	3	*HAT22*, *HB*-7, *HB40*
NINJA	3	*AFP1*, *AFP2*, *AFP3*
GATA	2	*GATA2*, *GATA12*
BELL	1	*BLH1*
C2H2 zinc finger	1	*ZFP5*
DRG	1	*AITR5*
FRS	1	*FAR1*
GARP	1	*BOA*
HSF	1	*HSFA6B*
KNOX	1	*KNAT3*
MADS	1	*AGL14*
NF-X	1	*NFXL1*
TGA	1	*RAS1*
WOX	1	*WOX13*

## Data Availability

The data presented in this study are available in the Appendix A. The data presented in this study are openly available in Gene Expression Omnibus (GEO); the GEO accession number is GSE229217.

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
