# Peer review of "Genome-Wide Analysis of Stress-Responsive Genes and Alternative Splice Variants in Arabidopsis Roots under Osmotic Stresses"

_ijms, 2023, doi:10.3390/ijms241914580_

Round 1
Reviewer 1 Report (New Reviewer)
In plants, the regulation of cell water content is regulated by water uptake in roots, but by stomata regulation of water loss in aerial. The current manuscript by Moon and colleagues describes an analysis of the transcription during osmotic shock in root and arial parts using RNA-seq and quantitative reverse transcription polymerase chain reaction. Specific emphasis is put on the analysis of 68 transcription factor genes. Interestingly, these do not include well-known transcription factors for ABA-dependent and independent genes, suggestions that further analysis of the identified factor genes may lead to new understanding of osmotic stress. Furthermore, the authors make predictions for the transcription factors binding to DNA response elements using a database of protein-DNA binding interaction. Overall, the systematic analysis of transcription during osmotic stress is well done and may provide a useful entry for more direct mechanistic studies. However, several issues need attention prior to publication.
Major issues
1. The analysis would be strengthened significantly by repeating after different times of stress as this would allow a deeper description of gene co-regulation.
2. The authors only report on genes with an absolute log2(fold change) ≥1. While these genes are the most interesting from the authors’ perspective, it would be useful for other investigators to also see results for genes with a lower absolute lfc. Please provide a table of lfc/FDR of all genes for which reproducible results were obtained.
3. Parts of the manuscript (e.g., lines) 36-83 are hard to read. Please, deconvolute such passages. Similarly, grammatic errors need attention.
4. Please indicate which proteins are encoded by the genes mention in the text (e.g., RAB18, KIN1, and RD29B in lines 46-47).
Minor issues
1. Indicate the total number of genes for which reproducible results were obtained.
2. Figure S9 must be redone. The fonts are too small to be read even after increasing the word file to 500% enlargement.
The manuscript has grammatical errors. Some parts are weaving and unnecessarily long. I suggest that the authors get some help improving the language
Author Response
Dear Editor and Reviewers, We appreciate constructive criticism from reviewers on our manuscript (ijms-2546419) entitled ‘Genome-wide analysis of stress-responsive genes and alternative splice variants in Arabidopsis roots under osmotic stresses’. We carefully considered the reviewers’ comments and revised the manuscript accordingly. The revised parts in the manuscript are marked using track changes. We also made point-by-point responses to the reviewers’ comments and explained reasons for unrevised parts (see below). We marked our ‘response’ in blue to make it easier for you to read (please refer to the attached PDF file). We greatly appreciate your consideration. Sincerely yours, Yong-Hwan Moon Professor Department of Molecular Biology Pusan National University 63 beon-gil 2, Busandaehak-ro, Geumjeong-gu Busan 46241, KOREA Tel : +82-51-510-2592, Fax : +82-51-513-9258 Email: moonyh@pusan.ac.kr In plants, the regulation of cell water content is regulated by water uptake in roots, but by stomata regulation of water loss in aerial. The current manuscript by Moon and colleagues describes an analysis of the transcription during osmotic shock in root and arial parts using RNA-seq and quantitative reverse transcription polymerase chain reaction. Specific emphasis is put on the analysis of 68 transcription factor genes. Interestingly, these do not include well-known transcription factors for ABA-dependent and independent genes, suggestions that further analysis of the identified factor genes may lead to new understanding of osmotic stress. Furthermore, the authors make predictions for the transcription factors binding to DNA response elements using a database of protein-DNA binding interaction. Overall, the systematic analysis of transcription during osmotic stress is well done and may provide a useful entry for more direct mechanistic studies. However, several issues need attention prior to publication. Major issues 1. The analysis would be strengthened significantly by repeating after different times of stress as this would allow a deeper description of gene co-regulation. (Response) We appreciate the reviewer’s valuable comment. In our current study, our rationale for combining RNA samples collected from 1 to 4 hours for mRNA sequencing was to investigate the collective molecular changes that occur during the early and late stress responses. We then validated the levels of selected genes from the mRNA-Seq analysis at different times of osmotic stress by using qRT-PCR. Conducting mRNA-Seq analysis for different stress time points would enhance our understanding of gene co-regulation. However, this work is both time-consuming and expensive. We would like to explore it in a future follow-up study. 2. The authors only report on genes with an absolute log2(fold change) ≥1. While these genes are the most interesting from the authors’ perspective, it would be useful for other investigators to also see results for genes with a lower absolute lfc. Please provide a table of lfc/FDR of all genes for which reproducible results were obtained. (Response) We have provided a new table (Table S1) that displays the fold change, FDR, and gene information for all genes with a lower absolute LFC/FDR. 3. Parts of the manuscript (e.g., lines) 36-83 are hard to read. Please, deconvolute such passages. Similarly, grammatic errors need attention. (Response) We apologize for the shortcomings. We have extensively revised Abstract, Introduction, and Discussion sections to improve the context and make them easier to read by deleting unnecessary sentences. In addition, the English in the manuscript has been further corrected by a team of English editing service (Editage: www.editage.co.kr). We have submitted the certificate of English editing service from Editage. 4. Please indicate which proteins are encoded by the genes mention in the text (e.g., RAB18, KIN1, and RD29B in lines 46-47). (Response) We have added protein information of RAB18, KIN1, and RD29B (line 48). Minor issues 1. Indicate the total number of genes for which reproducible results were obtained. (Response) The total number of genes that showed different expression levels with reproducibility is shown in the text (line 103–107) and Table 1. If our response to this comment is inadequate, we would appreciate it if the reviewer clarifies their concern. 2. Figure S9 must be redone. The fonts are too small to be read even after increasing the word file to 500% enlargement. (Response) We have enlarged Figure S9. Comments on the Quality of English Language The manuscript has grammatical errors. Some parts are weaving and unnecessarily long. I suggest that the authors get some help improving the language (Response) As we mentioned above, we made significant revisions to the Abstract, Introduction, and Discussion sections to enhance the context and improve readability. In addition, the English in the manuscript has been corrected through an English editing service team (Editage: www.editage.co.kr). We have submitted the certificate of English editing service from Editage.

Reviewer 2 Report (New Reviewer)
Authors performed an RNA-seq experiment on Arabidopsis, with response to drought and salinity stresses and presented a number of genes which changed its expression.
First of all, there are a number of papers on Arabidopsis (and other species also in IJMS, which I personally reviewed) investigating gene expression in response to drought. Authors should explain what is new in this work and how it is agreed with others.
Second, known ABA-dependent and -independent osmotic stress-responsive genes are not changed or changed only slightly. This is a strong indication of random nature of the presented results.
Gene expression changes constantly, also due to biological variance. It is not enough just to show the number of genes with altered expression, but the relevance to the biological process must be shown. In the present state the genes are assumed to be randomly changed and the relevance to the water stress is not shown.
In abstract and introduction there are many obvious things, which should be avoided:
Plant shoots and roots respond differently to osmotic stress; A no-treatment condition; etc
Number of genes on figure 1DEF is too small, which could be interpreted as statistical artifact. The only group with sufficient number of genes is a metabolic pathways. But the number of genes of metabolic pathway is around 30 on figure d and E and around 100 on figure f, that means mannitol works better alone, but not in combinations with salt. This is a very strange and not explainable from biological point of view. On this basis I conclude that the presented results are of random nature. Authors should look for explanatory models or other conformation and proof for the results.
Very contradictory results are, for example on figure 2: RAB18 shows no changes in expression, but in the text is described as differentially expressed.
We further identified 68 closely connected transcription factor genes that are involved in osmotic stress signal transduction - Where you show that?
Generally, the English is acceptable, except for some sentences which should be corrected, examples are below:
enriched KEGG mapper was selected by p <0.05
Load annotation data as Automatically download, and organism as Arabidopsis thaliana.
Author Response
Dear Editor and Reviewers,
We appreciate constructive criticism from reviewers on our manuscript (ijms-2546419) entitled ‘Genome-wide analysis of stress-responsive genes and alternative splice variants in Arabidopsis roots under osmotic stresses’. We carefully considered the reviewers’ comments and revised the manuscript accordingly. The revised parts in the manuscript are marked using track changes. We also made point-by-point responses to the reviewers’ comments and explained reasons for unrevised parts (see below). We marked our ‘response’ in blue to make it easier for you to read (please refer to the attached PDF file).
We greatly appreciate your consideration.
Sincerely yours,
Yong-Hwan Moon
Professor
Department of Molecular Biology
Pusan National University
63 beon-gil 2, Busandaehak-ro, Geumjeong-gu
Busan 46241, KOREA
Tel : +82-51-510-2592, Fax : +82-51-513-9258
Email: moonyh@pusan.ac.kr
<Reviewer 2>
Comments and Suggestions for Authors
Authors performed an RNA-seq experiment on Arabidopsis, with response to drought and salinity stresses and presented a number of genes which changed its expression.
First of all, there are a number of papers on Arabidopsis (and other species also in IJMS, which I personally reviewed) investigating gene expression in response to drought. Authors should explain what is new in this work and how it is agreed with others.
(Response) Thank you for the reviewer’s valuable feedback. In the original manuscript, we have discussed potential pathways that they may be involved (as described in lines 375–384). Although we want to understand more functions of newly identified genes and signaling pathways, there are limitations in the current approach. We aimed to conduct a functional analysis of the newly identified genes to uncover osmotic stress responses that are specific to roots in the future. Additionally, we compared our study with previous meta-analysis studies using microarray to understand possible signaling pathways, in which newly identified genes can be involved in. In previous studies, genes responsible for transcription factors have been found to be involved in osmotic stress in roots through various ways. These include cell wall modification, osmoprotectant synthesis and transport, reactive oxygen species (ROS) scavenging, protein metabolism, hormone signaling, and other mechanisms. Our study aligns with these findings.
Our manuscript is the first to report on genome-wide analysis data of osmotic stress response in Arabidopsis roots using mRNA-Seq. We also found that osmotic stress responses in roots might be mediated by uncharacterized pathways or mechanisms other than well-known ABA-dependent and/or ABA-independent osmotic stress-responsive genes. Furthermore, we identified 26 genes that showed varying expression of alternative splice variants or altered splicing events under salt and/or drought/osmotic stress conditions in Arabidopsis roots. We described these new findings in the Discussion.
Second, known ABA-dependent and -independent osmotic stress-responsive genes are not changed or changed only slightly. This is a strong indication of random nature of the presented results.
(Response) We disagree with the reviewer that our mRNA-Seq result is a result of random nature. We included qRT-PCR data of stress marker genes to validate the effectiveness of the stress treatments applied to the mRNA-Seq samples (Supplementary Figure S1). The data showed that the expression of RD29A and RAB18, which are the well-known salt and drought stress treatment marker genes, was significantly increased by salt or drought stress treatments. This suggests that our NaCl and mannitol treatment was performed properly. Additionally, we confirmed the suitability of mRNA-Seq data by quality control data such as mapping rate, alignment rate, etc. (Table S1). Through this process, we confirmed that our mRNA-Seq result is appropriate and acceptable to be analyzed to identify osmotic stress response in roots.
In this study, we want to identify osmotic stress responsive genes in roots. Expressions of well-known ABA-dependent and -independent osmotic stress-responsive genes were highly increased in whole seedlings under NaCl and mannitol treatment conditions in our mRNA-Seq result. However, expressions of those genes were lower or marginally different in roots compared to whole seedlings. These data indicate that response to osmotic stress in roots may be mediated by pathways or mechanisms that are not yet characterized and are distinct from well-known osmotic stress-responsive pathways.
Gene expression changes constantly, also due to biological variance. It is not enough just to show the number of genes with altered expression, but the relevance to the biological process must be shown. In the present state the genes are assumed to be randomly changed and the relevance to the water stress is not shown.
(Response) We thank the reviewer for pointing this out. Our Gene Ontology analysis in the biological process category showed that genes upregulated by NaCl treatment were predominantly associated with responding to abscisic acid, oxidative stress, and signal transduction. Additionally, genes upregulated by mannitol treatment were notably linked to responding to water deprivation, abscisic acid, salt stress, osmotic stress, and oxidative stress. These data suggest that changes in gene expression observed in root through mRNA-seq analysis are a consequence of salt or mannitol stress and carry biological significance. We have added revised sentences to better demonstrate the relevance of GO analysis (line 115–118; 120–123; 131–133).
In abstract and introduction there are many obvious things, which should be avoided:
Plant shoots and roots respond differently to osmotic stress; A no-treatment condition; etc
(Response) We extensively revised Abstract and Introduction to remove sentences for the obvious things.
Number of genes on figure 1DEF is too small, which could be interpreted as statistical artifact. The only group with sufficient number of genes is a metabolic pathways. But the number of genes of metabolic pathway is around 30 on figure d and E and around 100 on figure f, that means mannitol works better alone, but not in combinations with salt. This is a very strange and not explainable from biological point of view. On this basis I conclude that the presented results are of random nature. Authors should look for explanatory models or other conformation and proof for the results.
(Response) We thank the reviewer for this valuable comment. In the Figure 1DEF, we analyzed enriched KEGG pathways of upregulated genes and simply described the result in the text. In the revised manuscript, we revised the statement of KEGG pathway result to describe that the metabolic pathway is meaningful according to the reviewer’s suggestion (line 172–173). As we mentioned above, we confirmed that our mRNA-Seq result is appropriate and acceptable for analysis in terms of technical aspects and stress conditions (Figure S1 and Table S1).
Although NaCl or mannitol was treated, expression levels of genes can vary depending on specific treatment conditions, such as NaCl or mannitol concentration, the duration of treatment, the stage of the plant, and the growth conditions. As a result, there were differences in the numbers of up- and downregulated genes.
Very contradictory results are, for example on figure 2: RAB18 shows no changes in expression, but in the text is described as differentially expressed.
(Response) Fold changes of RAB18 are 38.091, 50.712, 117.821, and 216.075, in whole seedlings under NaCl treatment, whole seedlings under mannitol treatment, roots under NaCl treatment, and roots under mannitol treatment, respectively (Please see Table S2). Although the fold changes of RAB18 are different, their colors in Figure 2 appeared to be the same because the color range of over 10-fold is indistinguishable. To show the difference in expression among the samples more clearly, we have made a revision to Figure 2. The updated Figure 2 now displays a color range of 100-fold for the highest increase, effectively showing the differential expression.
We further identified 68 closely connected transcription factor genes that are involved in osmotic stress signal transduction - Where you show that?
(Response) We appreciate the reviewer’s comments. The 68 transcription factor genes were increased expression in roots under NaCl and mannitol treatment conditions (Table 2, Table S3, and Figure S6). In addition, these genes showed a close connection through protein network analysis using STRING, as shown in Figure 3. We described these in text (line 208–213) and showed them as Table 2 and Table S3.
Comments on the Quality of English Language
Generally, the English is acceptable, except for some sentences which should be corrected, examples are below:
enriched KEGG mapper was selected by p <0.05
Load annotation data as Automatically download, and organism as Arabidopsis thaliana.
(Response) We revised the sentences (line 161–162; 194; 290–291).

Reviewer 3 Report (New Reviewer)
Authors used RNA-seq technique to investigate responsive genes and mechanisms in Arabidopsis roots under osmotic stress. They showed that the changes occurred in the roots was different from the changes in the shoot. The study was well-designed and analyzed and they found various TF families involved in osmotic stress response of Arabidopsis. There is some missing information in MM section including; number of biological and technical samples had been used for RNA-seq, number of reads and raw data volume, length of reads….
Author Response
Dear Editor and Reviewers,
We appreciate constructive criticism from reviewers on our manuscript (ijms-2546419) entitled ‘Genome-wide analysis of stress-responsive genes and alternative splice variants in Arabidopsis roots under osmotic stresses’. We carefully considered the reviewers’ comments and revised the manuscript accordingly. The revised parts in the manuscript are marked using track changes. We also made point-by-point responses to the reviewers’ comments and explained reasons for unrevised parts (see below). We marked our ‘response’ in blue to make it easier for you to read (please refer to the attached PDF file).
We greatly appreciate your consideration.
Sincerely yours,
Yong-Hwan Moon
Professor
Department of Molecular Biology
Pusan National University
63 beon-gil 2, Busandaehak-ro, Geumjeong-gu
Busan 46241, KOREA
Tel : +82-51-510-2592, Fax : +82-51-513-9258
Email: moonyh@pusan.ac.kr
<Reviewer 3>
Authors used RNA-seq technique to investigate responsive genes and mechanisms in Arabidopsis roots under osmotic stress. They showed that the changes occurred in the roots was different from the changes in the shoot. The study was well-designed and analyzed and they found various TF families involved in osmotic stress response of Arabidopsis. There is some missing information in MM section including; number of biological and technical samples had been used for RNA-seq, number of reads and raw data volume, length of reads….
(Response) We added biological and technical numbers for RNA-Seq in Materials and methods (line 464–465). We also added RNA-Seq suitability information in Materials and methods (line 474–475).

Round 2
Reviewer 1 Report (New Reviewer)
Thank you for addressing my questions and comments. Regarding my Minor issue 1, I suggest that you indicate the total number of genes for which data was collected with reference to Table S2 (37,981).
Author Response
<Reviewer 1>
Thank you for addressing my questions and comments. Regarding my Minor issue 1, I suggest that you indicate the total number of genes for which data was collected with reference to Table S2 (37,981).
(Response) Thank you for your suggestion. We added sentences (lines 99-100) to indicate the total number of genes in Table S2.
(Revision details) Genes having very low abundance were removed from the analysis, leaving 37,980 genes for further analysis (Table S2).

Reviewer 2 Report (New Reviewer)
Simple search in Pubmed results in >400 papers on osmotic stress + roots + arabidopsis, with over 30 papers/year for the last 5 years. The novelty of the present paper is unclear. Few examples: PMID: 36833342, 33069038, 35954186, 32498390, ….
Second, simple RNA-seq plus GO, KEGG etc classifications is not enough. Experimental proof for candidate genes is a must, for example, overexpression, 2-hybrid arrays etc. Experimental verification of functional relevance of identified genes must be presented.
Moreover, to date responses to abiotic stresses are investigated in details, for specific gene groups etc, but not a simple - which genes changed their expression. On this basis the paper has no interest and presents no new results, and must be rejected.
Few comments on contradictions throughout the text:
Fig 2: RAB18 shows very small changes in expression relative to other genes. Moreover, changes between root and seedling are MUCH higher for all presented genes, compared to salt/mannitol treatments. Same on Fig1A. Therefore, changes after treatment are much less significant compared to expression differences between organs.
.Again: Gene expression changes constantly, also due to biological variance. It is not enough just to show the number of genes with altered expression, but the RELEVANCE to the biological process must be shown. Additional rtPCR only shows that gene expression is indeed changed, but why - was it random, or a response to other factors? I do not understand how “abscisic acid” is connected to NaCL, or why there is no “salt stress” after NaCL treatment. The paper is full of contradictions.
Author Response
<Reviewer 2>
Simple search in Pubmed results in >400 papers on osmotic stress + roots + arabidopsis, with over 30 papers/year for the last 5 years. The novelty of the present paper is unclear. Few examples: PMID: 36833342, 33069038, 35954186, 32498390, ….
(Response) As you indicated, there are many papers encompassing the keywords “osmotic stress + roots + Arabidopsis”. We also have published more than 12 articles about osmotic stress-responsive genes in Arabidopsis since 2008. However, these papers do not align with the specific objectives/goals, research interests, and arguments of our current study. We aimed to determine genome-widely whether root responses to osmotic stress are mediated by the established osmotic stress-responsive genes of the ABA-dependent and –independent pathways or if other uncharacterized signaling pathways are involved. According to mRNA-Seq and further analyses, response to osmotic stress in roots may be mediated by pathways or mechanisms that are not yet characterized and are distinct from well-known osmotic stress-responsive pathways. Moreover, our study includes the meta-analysis data, demonstrating that alternative splicing and alternative splice variants are important in the osmotic stress response in roots. Our manuscript is the first report on meta-analysis data of osmotic stress response in Arabidopsis roots using mRNA-Seq. We think that these are new findings and the novelty of our article, compared to other published articles. We believe that our study provides novel information and opens a new avenue of research on the understanding of osmotic stress response and regulatory mechanisms in this field.
About the reviewer’s specific examples, they fall outside the scope of our current report:
PMID 36833342: In this study, LEA family genes in Brassica campestris L. are analyzed. In addition, the function of BcLEA73 in salt and osmotic stress response is studied.
PMID 33069038: This study demonstrates that AtMYB109, an Arabidopsis MYB family gene, regulates stomatal closure by SA-mediated mechanism. AtMYB109 is also involved in pollen tube growth.
PMID 35954186: This study demonstrates that combined drought and heat stresses affect differential cell wall remodeling depending on developmental stage, resulting in a differential effect on suberization.
PMID 32498390: This study demonstrates that AtSK11 and AtSK12, protein kinase family in Arabidopsis, are involved in mild osmotic stress, especially in root growth. According to RNA-Seq analysis using WT and atsk11atsk12 mutants, bHLH69 may directly regulate the expression of mild osmotic stress-responsive genes.
Second, simple RNA-seq plus GO, KEGG etc classifications is not enough. Experimental proof for candidate genes is a must, for example, overexpression, 2-hybrid arrays etc. Experimental verification of functional relevance of identified genes must be presented.
Moreover, to date responses to abiotic stresses are investigated in details, for specific gene groups etc, but not a simple - which genes changed their expression. On this basis the paper has no interest and presents no new results, and must be rejected.
(Response) Thank you for your valuable suggestions. As we explained above, we believe that our study provides novel information and opens a new avenue of research on the understanding of osmotic stress response and regulatory mechanisms in this field. Our mRNA-Seq and additional analyses indicated that the response to osmotic stress in roots involves pathways and mechanisms that have not been fully characterized and differ from well-known stress-responsive pathways. Furthermore, our study includes the meta-analysis data, demonstrating that alternative splicing and alternative splice variants are important in the osmotic stress response in roots. Our manuscript represents the first meta-analysis of osmotic stress response in Arabidopsis roots using mRNA-seq data.
Although we want to understand more functions of newly identified genes and signaling pathways, there are limitations in the current approach. Functional analysis using overexpressing transgenic plants and/or knock-out mutants is required to understand them. We are performing a functional analysis of the newly identified genes to uncover osmotic stress responses and will report the results in other articles in the future.
Few comments on contradictions throughout the text:
Fig 2: RAB18 shows very small changes in expression relative to other genes. Moreover, changes between root and seedling are MUCH higher for all presented genes, compared to salt/mannitol treatments. Same on Fig1A. Therefore, changes after treatment are much less significant compared to expression differences between organs.
(Response) We believe there might have been a misunderstanding regarding our data in Figure 2 by the reviewer. Fig. 2 (and Table S3) shows the relative expression changes (fold changes) in whole seedlings and roots under NaCl and mannitol treatments. For example, the fold changes in the first column were obtained by comparing the expression levels between the control whole seedlings (no treatment condition) and NaCl-treated whole seedlings. It is important to note that the comparison of gene fold changes between whole seedlings and roots, as the reviewer mentioned in relation to Fig. 1 and 2, is not feasible. We have revised the sentence (line 494) in the Materials and Methods section to clarify the data Fig. 2 and Table S3.
In Fig. 2 and Table S3, the expression of the well-known ABA-dependent and ABA-independent osmotic stress-responsive genes in whole seedlings was significantly increased by NaCl and mannitol treatments as expected. However, in roots, the expression of certain genes did not show significant changes with NaCl and mannitol treatments, suggesting that the response to osmotic stress in roots may involve pathways and mechanisms that differ from the well-known stress-responsive pathways. These results are one of new findings in our manuscript.
Again: Gene expression changes constantly, also due to biological variance. It is not enough just to show the number of genes with altered expression, but the RELEVANCE to the biological process must be shown. Additional rtPCR only shows that gene expression is indeed changed, but why - was it random, or a response to other factors? I do not understand how “abscisic acid” is connected to NaCL, or why there is no “salt stress” after NaCL treatment. The paper is full of contradictions.
(Response) The NaCl and mannitol treatment methods used in this study are not unique to our manuscript; rather, they are generally accepted and well-established in the field of stress research. Using qRT-PCR, we checked the expression of the osmotic stress marker genes, such as RD29A and RAB18, in samples treated with NaCl or mannitol at different time points in order to validate the effectiveness of the stress treatments (Fig. 4, Fig. 5, and Fig. S1). The results indicate that NaCl and mannitol were effectively applied to the samples, and the samples responded to salt and drought stress, respectively. Therefore, we concluded that the expression patterns of the selected genes resulted from the treatments. In addition, mRNA-Seq results demonstrate that the expression of certain well-known ABA-dependent and ABA-independent stress-responsive genes is significantly upregulated under NaCl and mannitol treatments (Figure 2 and Table S3), verifying that our NaCl and mannitol treatments are proper to induce salt and drought stress conditions.
We agree with the reviewer that the biological and functional roles of the selected genes in the osmotic stress response should be analyzed for an in-depth understanding of more functions of newly identified genes and signaling pathways. We are currently performing functional analysis of a few selected genes using overexpressing transgenic plants and/or knock-out mutants. We plan to report the results in separate papers.
Abscisic acid (ABA), a phytohormone, is well known to be involved in stress response, especially osmotic stress response in plants (Trends Plant Sci (2018) 23, 513–522; Curr Opin Plant Biol (2014) 21, 133–139; Plant Cell Rep (2013) 32, 971–983; J Plant Res (2011) 124, 509–525). Thus, it is reasonable that “response to abscisic acid” is shown in biological processes under NaCl and mannitol treatments.
While we were confirming our Gene Ontology (GO) enrichment analysis data in Fig. S3 during the revision period, we found that upregulated genes under both NaCl and mannitol treatments in roots (361 genes in Fig. 1b) were not included in the analysis. Consequently, we re-analyzed GO enrichment by including these genes and then revised Fig. S3 and the main text (lines 117-135) accordingly. The revised results indicate that NaCl treatment in roots is associated with a “response to salt stress” in the biological process category. Moreover, we observed osmotic stress-related GO terms, such as response to water deprivation, response to abscisic acid, response to oxidative stress, and response to cold. These updated data provide support for the proper execution of our mRNA-Seq experiment including NaCl and mannitol treatments and confirm that our mRNA-Seq data is acceptable for analysis.
As we explained above, we do not think that our manuscript includes a contradiction.

This manuscript is a resubmission of an earlier submission. The following is a list of the peer review reports and author responses from that submission.
Round 1
Reviewer 1 Report
Please see attached for my detailed comments.

English writing is good in general
Author Response
Dear Editor and Reviewers,
We appreciate constructive feedback from the editor and the reviewers on our manuscript (ijms-2445482) entitled ‘Genome-wide analysis of stress-responsive genes and alternative splice variants in Arabidopsis roots under osmotic stresses’. We agree with most of the comments and revised the manuscript accordingly. Revised parts in the manuscript are marked using track changes. We also made point-by-point responses to the reviewers’ comments and explained reasons for unrevised parts.
We greatly appreciate your consideration.
Sincerely yours,
Yong-Hwan Moon
Professor
Department of Molecular Biology
Pusan National University
63 beon-gil 2, Busandaehak-ro, Geumjeong-gu
Busan 46241, KOREA
Tel : +82-51-510-2592, Fax : +82-51-513-9258
Email: moonyh@pusan.ac.kr
<Reviewer 1>
In this submission, the authors properly addressed most of my comments from last round and revised the manuscript accordingly. Below are my remaining questions.
Major
- According to authors’ response, all the p-values reported in this study were unadjusted/uncorrected raw p-values. As a common practice for DEG analysis, please correct all the p-values with metrics like FDR or BH correction and update all the results and conclusions accordingly.
(response) We corrected the p-values to FDR for DEG analysis as suggested by the reviewer. Consequently, we have made revisions to the relevant figures and tables (Table 1, Figure 1, figure S2, Figure S3, and Figure S4)
- Regarding the missing qPCR validation for the mannitol treatment, there is no analysis presented in the current manuscript to directly show that ‘most of the selected genes showed similar expression levels under salt and drought stress conditions. Besides, there is no guarantee that the qPCR results for these genes will be similar for the salt and drought conditions. It is recommended to add qPCR validation for the mannitol treatment for the consistency and completion of the manuscript.
(response) We performed RT-qPCR to validate mannitol treatment and included the results in Figure 5, Figure 8, and Figure S7. In Figure 5 and Figure S7, the expression of AT1G62975, AITR5, WRKY29, BOA, and MYB3 was significantly increased under mannitol treatment conditions, confirming the validity of the mRNA-Seq analysis.
Minor
- The node colors in Figure 3 are still confusing. Do they represent p-value or fold change? Please be more specific and add a color bar to show the range.
(response) We revised the legend for Figure 3 to provide more detailed information and included a color scale bar to show the range.
Therefore, I cannot condone publication before these questions have been addressed.
Reviewer 2 Report
The manuscript has been significantly improved, which can be accepted in present form.
The language can be carefully edited by a native English speaker with strong background in plant science.
Author Response
Dear Editor and Reviewers,
We appreciate constructive feedback from the editor and the reviewers on our manuscript (ijms-2445482) entitled ‘Genome-wide analysis of stress-responsive genes and alternative splice variants in Arabidopsis roots under osmotic stresses’. We agree with most of the comments and revised the manuscript accordingly. Revised parts in the manuscript are marked using track changes. We also made point-by-point responses to the reviewers’ comments and explained reasons for unrevised parts.
We greatly appreciate your consideration.
Sincerely yours,
Yong-Hwan Moon
Professor
Department of Molecular Biology
Pusan National University
63 beon-gil 2, Busandaehak-ro, Geumjeong-gu
Busan 46241, KOREA
Tel : +82-51-510-2592, Fax : +82-51-513-9258
Email: moonyh@pusan.ac.kr
<Reviewer 2>
The manuscript has been significantly improved, which can be accepted in present form.
(response) We thank the reviewer for the positive comments.
Reviewer 3 Report
Although the authors made some corrections in the manuscript, several previous comments were ignored especially later to the significance of the study. Moreover, the discussion should be strengthened by providing a holistic understanding of the roles of the identified transcription factors and their downstream signaling pathways in stress responses. Overall, the manuscript does not offer novel findings at the end of RNA-seq while emphasizing the alternative splice variants may shape the manuscript in a different direction so that it can become slightly differ from the literature.
Many sentences are corrected now.
Author Response
Dear Editor and Reviewers,
We appreciate constructive feedback from the editor and the reviewers on our manuscript (ijms-2445482) entitled ‘Genome-wide analysis of stress-responsive genes and alternative splice variants in Arabidopsis roots under osmotic stresses’. We agree with most of the comments and revised the manuscript accordingly. Revised parts in the manuscript are marked using track changes. We also made point-by-point responses to the reviewers’ comments and explained reasons for unrevised parts.
We greatly appreciate your consideration.
Sincerely yours,
Yong-Hwan Moon
Professor
Department of Molecular Biology
Pusan National University
63 beon-gil 2, Busandaehak-ro, Geumjeong-gu
Busan 46241, KOREA
Tel : +82-51-510-2592, Fax : +82-51-513-9258
Email: moonyh@pusan.ac.kr
<Reviewer 3>
Although the authors made some corrections in the manuscript, several previous comments were ignored especially later to the significance of the study. Moreover, the discussion should be strengthened by providing a holistic understanding of the roles of the identified transcription factors and their downstream signaling pathways in stress responses. Overall, the manuscript does not offer novel findings at the end of RNA-seq while emphasizing the alternative splice variants may shape the manuscript in a different direction so that it can become slightly differ from the literature.
(response) Thank you for your valuable feedback. Actually, we had revised the Results (Figures) and Materials and methods sections based on your previous comments. Additionally, we have provided a detailed description of the roles of the identified transcription factors such as MYB71, MYB108, MYB12, and AITR5, and their potential involvement in osmotic stress responses in the Discussion of the resubmitted manuscript.
In the current revised manuscript, we have additionally discussed potential pathways that they may be involved (as described in lines 396–405). Although we want to understand more functions of newly identified genes and signaling pathways, there are limitations in the current approach. Functional analysis using overexpressing transgenic plants and/or knock-out mutants is required to understand them. We aim to conduct a functional analysis of the newly identified genes to uncover osmotic stress responses that are specific to roots in the future.
Our manuscript is the first to report on genome-wide analysis data of osmotic stress response in Arabidopsis roots using mRNA-Seq. We also found that osmotic stress responses in roots might be mediated by uncharacterized pathways or mechanisms other than well-known ABA-dependent and/or ABA-independent osmotic stress-responsive genes. Furthermore, we identified 26 genes that showed varying expression of alternative splice variants or altered splicing events under salt and/or drought/osmotic stress conditions in Arabidopsis roots
We described our new findings in the Discussion.
Reviewer 4 Report
Scientific comments:
- Please italicize the species name throughout the document.
- A better definition of the aims is needed. The authors expressed that "we performed transcriptional profiling of Arabidopsis roots under osmotic stress conditions such as high salinity and drought/osmotic stress using NGS to assess gene expression changes in the roots of Arabidopsis and therefore elucidate the molecular mechanisms underlying the osmotic stress response in roots." This is quite general and very technological. What is the hypothesis being tested here or what did the authors expect to obtain? This needs to be clear.
- Many methodological details are missing and I cannot fully validate the results without them. For instance, the number of biological or technical replicates needs to be included in M&Ms. Without this it is not even possible to address if the bioinformatic packages used are indeed accurate.
- A major point is the definition of DEGs. The authors stated as ". Genes with > 2- fold differences in expression with p < 0.05 were considered upregulated or downregulated". However, the expression has many definitions and this needs to be clear. In addition, up or down- in relation to what? Obviously the control...but it needs to be indicated.
- The programs used are indicated without any definitions. This is crucial. For instance, the network retrieved from STRING results is *highly* dependent on the query made. That is not indicated. The same comment occurs in other packages.
- In relation to results, authors should clarify several sections. For instance: "Genes with > 2-fold differences in expression 121 with p < 0.05 were considered up- or down-regulated under salt and/or drought/osmotic 122 stress conditions". This type of definition is written many times in the document. However, salinity also imposes an osmotic stress.
- No statistical results of RNA sequencing are indicated in the main document.
- Several figures have the names in the axis incompleted.
- How exactly were the ABA-dependent and -independent osmotic stress-responsive genes selected? I found this puzzling since some genes act in ABA- related and also not-related pathways...
- In relation to rt-PCR, the authors indicated "Two technical replicates were performed for each biological replicate. At least two biological replicates showed similar results, with one shown here. Error bars represent standard deviation (n = 6 reactions)." This is simply wrong; the authors cannot choose the replicate they want. That is the reason for having biological and technical replicates....
- The discussion needs to be updated. There are many genomic studies showing the impacts of osmotic stress in plants, including also Arabidopsis.
Proofreading comments:
- The authors should carefully revise the text (including the abstract and the title). Many sentences have typos and grammatical mistakes, as well as repetitive words. That occurs throughout the text.
Proofreading comments:
- The authors should carefully revise the text (including the abstract and the title). Many sentences have typos and grammatical mistakes, as well as repetitive words. That occurs throughout the text.
Author Response
Dear Editor and Reviewers,
We appreciate constructive feedback from the editor and the reviewers on our manuscript (ijms-2445482) entitled ‘Genome-wide analysis of stress-responsive genes and alternative splice variants in Arabidopsis roots under osmotic stresses’. We agree with most of the comments and revised the manuscript accordingly. Revised parts in the manuscript are marked using track changes. We also made point-by-point responses to the reviewers’ comments and explained reasons for unrevised parts.
We greatly appreciate your consideration.
Sincerely yours,
Yong-Hwan Moon
Professor
Department of Molecular Biology
Pusan National University
63 beon-gil 2, Busandaehak-ro, Geumjeong-gu
Busan 46241, KOREA
Tel : +82-51-510-2592, Fax : +82-51-513-9258
Email: moonyh@pusan.ac.kr
<Reviewer 4>
Scientific comments:
- Please italicize the species name throughout the document.
(response) We have checked and confirmed that all species names are in italics in our current manuscript. We speculate that the italics may be converted to plain text in different versions of MS Word. Furthermore, Arabidopsis can be formatted as either plain or italics in IJMS.
- A better definition of the aims is needed. The authors expressed that "we performed transcriptional profiling of Arabidopsis roots under osmotic stress conditions such as high salinity and drought/osmotic stress using NGS to assess gene expression changes in the roots of Arabidopsis and therefore elucidate the molecular mechanisms underlying the osmotic stress response in roots." This is quite general and very technological. What is the hypothesis being tested here or what did the authors expect to obtain? This needs to be clear.
(response) Our study aims to analyze the genome-wide gene expression changes in roots under osmotic stress conditions to identify root-specific genes and mechanisms that respond to osmotic stress. We have clarified the purpose of our study in the revised manuscript.
- Many methodological details are missing and I cannot fully validate the results without them. For instance, the number of biological or technical replicates needs to be included in M&Ms. Without this it is not even possible to address if the bioinformatic packages used are indeed accurate.
(response) We added the number of biological or technical replicates in the materials and methods.
- A major point is the definition of DEGs. The authors stated as ". Genes with > 2- fold differences in expression with p < 0.05 were considered upregulated or downregulated". However, the expression has many definitions and this needs to be clear. In addition, up or down- in relation to what? Obviously the control...but it needs to be indicated.
(response) We added descriptions to clarify the upregulated and downregulated genes in both the results and the materials and methods sections.
- The programs used are indicated without any definitions. This is crucial. For instance, the network retrieved from STRING results is *highly* dependent on the query made. That is not indicated. The same comment occurs in other packages.
(response) We revised the legend for Figure 3 to provide additional details.
- In relation to results, authors should clarify several sections. For instance: "Genes with > 2-fold differences in expression 121 with p < 0.05 were considered up- or down-regulated under salt and/or drought/osmotic 122 stress conditions". This type of definition is written many times in the document. However, salinity also imposes an osmotic stress.
(response) In a previous review, it was commented that mannitol treatment induces both osmotic and drought stress. Thus, we consider and define mannitol treatment as a form of drought/osmotic stress. In addition, salt treatment can lead to ionic toxicity and osmotic stress.
- No statistical results of RNA sequencing are indicated in the main document.
(response) We added statistical results, FDR, in Table S2, Table S3, and Table S4.
- Several figures have the names in the axis incompleted.
(response) We revised axis in Figure 1 to show the names completely.
- How exactly were the ABA-dependent and -independent osmotic stress-responsive genes selected? I found this puzzling since some genes act in ABA- related and also not-related pathways...
(response) We reviewed published papers (Plant Cell (2016) 28, 2178–2196; Plant Cell (2001) 13, 2063–2083; Plant Cell Physiol (2015) 56, 930–942; Plant Cell (1998) 10, 1391–1406; Plant J (2010) 61, 1041–1052; Curr Opin Plant Biol (2014) 21, 133–139; Plant Cell (2008) 20, 1879–1898; Plant J (2007) 49, 184–193; Plant Mol Biol (2006) 61, 95–109) to identify ABA-dependent and –independent osmotic stress-responsive genes.
- In relation to rt-PCR, the authors indicated "Two technical replicates were performed for each biological replicate. At least two biological replicates showed similar results, with one shown here. Error bars represent standard deviation (n = 6 reactions)." This is simply wrong; the authors cannot choose the replicate they want. That is the reason for having biological and technical replicates....
(response) We reanalyzed our RT-qPCR results, those shown in Figures 4 and 7, by incorporating all of our biological and technical replicates.
- The discussion needs to be updated. There are many genomic studies showing the impacts of osmotic stress in plants, including also Arabidopsis.
(response) We discussed more about our findings compared to previous meta-analysis studies.
Proofreading comments:
- The authors should carefully revise the text (including the abstract and the title). Many sentences have typos and grammatical mistakes, as well as repetitive words. That occurs throughout the text.
(response) We had corrected grammatical and/or punctuation issues in the original manuscript (ijms-2360949) and its resubmitted manuscript (current manuscript; ijms-2445482) through English editing service (Editage).
Round 2
Reviewer 1 Report
All my comments and questions have been properly addressed.
English writing is clear in general
Author Response
We thank the reviewer for the positive comments.
Reviewer 3 Report
I thank the authors for elaborating on the discussion section.
Author Response

(The authors gave the same response as above.)

Reviewer 4 Report
Although the authors are replying that “We agree with most of the comments and revised the manuscript accordingly.”, this is simply not true. The authors have not addressed the concerns raised. The revision has poorly addressed the comments, which were raised to help the authors produce a better manuscript and to clarify concepts that were not clear to any reader. These flaws still occur in the revised version. For instance:
Previous comment #1: “Please italicize the species name throughout the document”
Authors response: “We have checked and confirmed that all species names are in italics in our current manuscript. We speculate that the italics may be converted to plain text in different versions of MS Word. Furthermore, Arabidopsis can be formatted as either plain or italics in IJMS.”
My current answer: nothing has changed and the statement of the authors that Arabidopsis can be formatted as plain in IJMS only has support in some specific cases, especially when it is used as a common name – it is not the case here.
Previous comment #2: “A better definition of the aims is needed. The authors expressed that "we performed transcriptional profiling of Arabidopsis roots under osmotic stress conditions such as high salinity and drought/osmotic stress using NGS to assess gene expression changes in the roots of Arabidopsis and therefore elucidate the molecular mechanisms underlying the osmotic stress response in roots." This is quite general and very technological. What is the hypothesis being tested here or what did the authors expect to obtain? This needs to be clear.”
Authors response: “Our study aims to analyze the genome-wide gene expression changes in roots under osmotic stress conditions to identify root-specific genes and mechanisms that respond to osmotic stress. We have clarified the purpose of our study in the revised manuscript.”
My current answer: What the authors have written in the manuscript is: “Given that there are distinct osmotic stress responses in shoots and roots, it suggests 114 that there are the genes responsive to osmotic stresses expressed specifically in the roots. 115 We, therefore, anticipated that genome-wide analysis under osmotic stress conditions in 116 the roots will contribute to identify novel osmotic stress-responsive genes and osmotic 117 stress-responsive mechanisms in the roots.” I appreciate the answer, but my previous comment still holds as the authors are basically giving the same answer as before. There are no specific mechanisms stated or hypothesis of what the authors expect to find. Of course that under stress there are always genes and pathways that are going to be changed..
Previous comment #3: “Many methodological details are missing and I cannot fully validate the results without them. For instance, the number of biological or technical replicates needs to be included in M&Ms. Without this it is not even possible to address if the bioinformatic packages used are indeed accurate.”
Authors response: “We added the number of biological or technical replicates in the materials and methods.”
My current answer: This comment still holds as no details have been added. In relation to the replicates, the authors only added a sentence in rt-PCR (nothing is added in the other sections). In relation to the rt-PCR the authors now wrote “At least two biological replicates were performed with two technical replicate for each biological replicate. Three independent reactions were performed for each technical replicate.” This implies a very low number of biological replicates and I have no idea what the authors mean here by technical replicates as they have mentioned two that were replicated (again?) three times…
The authors have also now added in some figures that “Transcript levels at 0 h in isoform 1 were set as 1.” The reason beyond this is not stated.
Previous comment #4: “A major point is the definition of DEGs. The authors stated as ". Genes with > 2- fold differences in expression with p < 0.05 were considered upregulated or downregulated". However, the expression has many definitions and this needs to be clear. In addition, up or down- in relation to what? Obviously the control...but it needs to be indicated.”
Authors response: “We added descriptions to clarify the upregulated and downregulated genes in both the results and the materials and methods sections.”
My current answer: The authors have written the control but they have not stated what is the control conditions, which seems to change across results/figures.
Previous comment #5: “The programs used are indicated without any definitions. This is crucial. For instance, the network retrieved from STRING results is *highly* dependent on the query made. That is not indicated. The same comment occurs in other packages.”
Authors response: “We revised the legend for Figure 3 to provide additional details.”
My current answer: Nothing has changed. The legend does not provide additional details allowing replication of this analysis. The authors have not added any details in the other programs. As a result, a huge proportion of this study cannot be replicated.
Previous comment #6: “In relation to results, authors should clarify several sections. For instance: "Genes with > 2-fold differences in expression 121 with p < 0.05 were considered up- or down-regulated under salt and/or drought/osmotic 122 stress conditions". This type of definition is written many times in the document. However, salinity also imposes an osmotic stress.”
Authors response: “In a previous review, it was commented that mannitol treatment induces both osmotic and drought stress. Thus, we consider and define mannitol treatment as a form of drought/osmotic stress. In addition, salt treatment can lead to ionic toxicity and osmotic stress.”
My current answer: Yes, this is what I am saying. And because of this many sentences are not easy to follow since the two treatments can induce osmotic stress, so the reader simply do not understand if the authors are mentioning the drought or the mannitol treatment.
Previous comment #7: “No statistical results of RNA sequencing are indicated in the main document.”
Authors response: “We added statistical results, FDR, in Table S2, Table S3, and Table S4.”
My current answer: This makes no sense. What do the authors mean by left reads and right reads?? They should have mapped a consensus…
Many figures and tables in these new SI files are incomplete and lack information on how they were obtained.
In addition: the authors are saying that all data are in SI files when obviously raw data should be deposited in a public repository.
Previous comment #8: “Several figures have the names in the axis incompleted.”
Authors response: “We revised axis in Figure 1 to show the names completely.”
My current answer: Thank you. But the same still holds for SI files.
Previous comment #9: “How exactly were the ABA-dependent and -independent osmotic stress-responsive genes selected? I found this puzzling since some genes act in ABA- related and also not-related pathways..”
Authors response: “We reviewed published papers (Plant Cell (2016) 28, 2178–2196; Plant Cell (2001) 13, 2063–2083; Plant Cell Physiol (2015) 56, 930–942; Plant Cell (1998) 10, 1391–1406; Plant J (2010) 61, 1041–1052; Curr Opin Plant Biol (2014) 21, 133–139; Plant Cell (2008) 20, 1879–1898; Plant J (2007) 49, 184–193; Plant Mol Biol (2006) 61, 95–109) to identify ABA-dependent and –independent osmotic stress-responsive genes.”
My current answer: My concern still holds as the reader has no idea how the authors choose those genes (not stated anywhere).
Previous comment #10: “In relation to rt-PCR, the authors indicated "Two technical replicates were performed for each biological replicate. At least two biological replicates showed similar results, with one shown here. Error bars represent standard deviation (n = 6 reactions)." This is simply wrong; the authors cannot choose the replicate they want. That is the reason for having biological and technical replicates....”
Authors response: “We reanalyzed our RT-qPCR results, those shown in Figures 4 and 7, by incorporating all of our biological and technical replicates.”
My current answer: See previous comment concerning biological and technical replicates.
Previous comment #11: “The discussion needs to be updated. There are many genomic studies showing the impacts of osmotic stress in plants, including also Arabidopsis.”
Authors response: “We discussed more about our findings compared to previous meta-analysis studies.”
My current answer: I still have the same concern. The authors have literally added 2 new sentences and as a result the discussion is still very poor. For instance, in relation to the authors aims, which has to find new genes and pathways acting in response to stress in roots of Arabidopsis what are the advances made by this study? How do this connect with other studies?
Previous comment #12: “Proofreading comments: The authors should carefully revise the text (including the abstract and the title). Many sentences have typos and grammatical mistakes, as well as repetitive words. That occurs throughout the text.”
Authors response: “We had corrected grammatical and/or punctuation issues in the original manuscript (ijms-2360949) and its resubmitted manuscript (current manuscript; ijms-2445482) through English editing service (Editage).”
My current answer: my concern still holds. I’m sorry but this is simply not true. Take a look into the new aims written, as an example: “Given that there are distinct osmotic stress responses in shoots and roots, it suggests that there are the genes responsive to osmotic stresses expressed specifically in the roots.”… This type of situation occurs in many other sentences in the manuscript.
I am sorry for not being more positive but there are too many things here that have no support and are not clear to any reader. There are also many details missing that prevent the replication of this study. As a result, I cannot recommend this version for publication, either here or in any other journal.
See above.
Author Response
Dear Editor and Reviewer,
We appreciate constructive criticism from the reviewer on our manuscript (ijms-2445482) entitled ‘Genome-wide analysis of stress-responsive genes and alternative splice variants in Arabidopsis roots under osmotic stresses’. We carefully considered the reviewer’s comments and revised the manuscript accordingly to ensure the reader’s understanding. The revised parts in the manuscript are marked using track changes. We also made point-by-point responses to the reviewer’s comments and explained reasons for unrevised parts (see below). We made an effort to address the reviewer’s comments. However, if there are still unrevised parts, we would appreciate it if the reviewer specifies them
We greatly appreciate your consideration.
Sincerely yours,
Yong-Hwan Moon
Professor
Department of Molecular Biology
Pusan National University
63 beon-gil 2, Busandaehak-ro, Geumjeong-gu
Busan 46241, KOREA
Tel : +82-51-510-2592, Fax : +82-51-513-9258
Email: moonyh@pusan.ac.kr
<Reviewer 4>
Although the authors are replying that “We agree with most of the comments and revised the manuscript accordingly.”, this is simply not true. The authors have not addressed the concerns raised. The revision has poorly addressed the comments, which were raised to help the authors produce a better manuscript and to clarify concepts that were not clear to any reader. These flaws still occur in the revised version. For instance:
Previous comment #1: “Please italicize the species name throughout the document”
Authors response: “We have checked and confirmed that all species names are in italics in our current manuscript. We speculate that the italics may be converted to plain text in different versions of MS Word. Furthermore, Arabidopsis can be formatted as either plain or italics in IJMS.”
My current answer: nothing has changed and the statement of the authors that Arabidopsis can be formatted as plain in IJMS only has support in some specific cases, especially when it is used as a common name – it is not the case here.
(response) We italicized “Arabidopsis” throughout the document. If there are any missed parts, we would appreciate it if the reviewer specifies them.
Previous comment #2: “A better definition of the aims is needed. The authors expressed that "we performed transcriptional profiling of Arabidopsis roots under osmotic stress conditions such as high salinity and drought/osmotic stress using NGS to assess gene expression changes in the roots of Arabidopsis and therefore elucidate the molecular mechanisms underlying the osmotic stress response in roots." This is quite general and very technological. What is the hypothesis being tested here or what did the authors expect to obtain? This needs to be clear.”
Authors response: “Our study aims to analyze the genome-wide gene expression changes in roots under osmotic stress conditions to identify root-specific genes and mechanisms that respond to osmotic stress. We have clarified the purpose of our study in the revised manuscript.”
My current answer: What the authors have written in the manuscript is: “Given that there are distinct osmotic stress responses in shoots and roots, it suggests 114 that there are the genes responsive to osmotic stresses expressed specifically in the roots. 115 We, therefore, anticipated that genome-wide analysis under osmotic stress conditions in 116 the roots will contribute to identify novel osmotic stress-responsive genes and osmotic 117 stress-responsive mechanisms in the roots.” I appreciate the answer, but my previous comment still holds as the authors are basically giving the same answer as before. There are no specific mechanisms stated or hypothesis of what the authors expect to find. Of course that under stress there are always genes and pathways that are going to be changed..
(response) Thank you for your feedback. We have added a statement outlining the purpose of our study in Introduction section.
(revision details)
“We aimed to determine whether the established osmotic stress-responsive genes, ABA-dependent and –independent, play a role in root responses to osmotic stress or if other uncharacterized signaling pathways are involved. Additionally, alternative splicing patterns and the expression of splicing factors may differ between roots and shoots.”
Previous comment #3: “Many methodological details are missing and I cannot fully validate the results without them. For instance, the number of biological or technical replicates needs to be included in M&Ms. Without this it is not even possible to address if the bioinformatic packages used are indeed accurate.”
Authors response: “We added the number of biological or technical replicates in the materials and methods.”
My current answer: This comment still holds as no details have been added. In relation to the replicates, the authors only added a sentence in rt-PCR (nothing is added in the other sections). In relation to the rt-PCR the authors now wrote “At least two biological replicates were performed with two technical replicate for each biological replicate. Three independent reactions were performed for each technical replicate.” This implies a very low number of biological replicates and I have no idea what the authors mean here by technical replicates as they have mentioned two that were replicated (again?) three times…
The authors have also now added in some figures that “Transcript levels at 0 h in isoform 1 were set as 1.” The reason beyond this is not stated.
(response) We have added the number of technical replicates and biological replicates in each figure legend (Figures 4, 5, 7, and 8). The data presented in Figure 5 and Figure 8 were generated in response to a reviewer’s request, and we performed two biological replicates due to a short revision timeline. We believe that the data presented in Figures 5 and 8 are consistent with mRNA-Seq data (Tables S3 and S4, Figure S6), and that these data mutually support each other.
We have added the reason (the representative model) for setting isoform 1 as a control in each relevant Figure legend (Figures 7 and 8).
(revision details)
Figure 4: Three biological replicates were performed with two technical replicates for each biological replicate. Three independent reactions were performed for each technical replicate.
Figure 5: Two biological replicates were performed with two technical replicates for each biological replicate. Three independent reactions were performed for each technical replicate.
Figure 7: Transcript levels at 0 h in isoform 1, the representative model (TAIR, https://www.arabidopsis.org/), were set as 1. Three biological replicates were performed with two technical replicates for each biological replicate. Three independent reactions were performed for each technical replicate.
Figure 8: Transcript levels at 0 h in isoform 1, the representative model (TAIR, https://www.arabidopsis.org/), were set as 1. Two biological replicates were performed with two technical replicates for each biological replicate. Three independent reactions were performed for each technical replicate.
Previous comment #4: “A major point is the definition of DEGs. The authors stated as ". Genes with > 2- fold differences in expression with p < 0.05 were considered upregulated or downregulated". However, the expression has many definitions and this needs to be clear. In addition, up or down- in relation to what? Obviously the control...but it needs to be indicated.”
Authors response: “We added descriptions to clarify the upregulated and downregulated genes in both the results and the materials and methods sections.”
My current answer: The authors have written the control but they have not stated what is the control conditions, which seems to change across results/figures.
(response) We have explained each control condition in the main text (lines 129, 467, and 521).
(revision details)
Line 129: A no-treatment condition (no NaCl or mannitol) was used as a control.
Line 467: Seedlings, shoots, or roots harvested at 0 h were used as controls.
Line 521: A no-treatment condition (no NaCl or mannitol) was used as a control.
Previous comment #5: “The programs used are indicated without any definitions. This is crucial. For instance, the network retrieved from STRING results is *highly* dependent on the query made. That is not indicated. The same comment occurs in other packages.”
Authors response: “We revised the legend for Figure 3 to provide additional details.”
My current answer: Nothing has changed. The legend does not provide additional details allowing replication of this analysis. The authors have not added any details in the other programs. As a result, a huge proportion of this study cannot be replicated.
(response) We have added the additional details in Figure 3 legend based on other published papers presenting the STRING results (Plant J (2019) 99, 176–194; Heliyon (2019) 5, e02614; PLoS One (2013) 8, e77261).
(revision details)
Figure 3. Protein network of the transcription factor families of upregulated genes in roots. Protein network interaction of the transcription factor genes was analyzed using STRING Apps of Cytoscape. STRING analysis was performed using protein query with gene symbols as identifiers, confidence score of 0.4, and maximum additional interactors of 0. Node color and size were set as fold change and FDR, respectively. Edge thickness represents the confidence in association between two connected nodes and ranges from 0.4 to 1.0, as determined by STRING.
Previous comment #6: “In relation to results, authors should clarify several sections. For instance: "Genes with > 2-fold differences in expression 121 with p < 0.05 were considered up- or down-regulated under salt and/or drought/osmotic 122 stress conditions". This type of definition is written many times in the document. However, salinity also imposes an osmotic stress.”
Authors response: “In a previous review, it was commented that mannitol treatment induces both osmotic and drought stress. Thus, we consider and define mannitol treatment as a form of drought/osmotic stress. In addition, salt treatment can lead to ionic toxicity and osmotic stress.”
My current answer: Yes, this is what I am saying. And because of this many sentences are not easy to follow since the two treatments can induce osmotic stress, so the reader simply do not understand if the authors are mentioning the drought or the mannitol treatment.
(response) In early part of Results section, we clarified the stress conditions that were used in our study (line 128), stating that we treated NaCl and mannitol for salt and drought stresses, respectively (Plant Cell Physiol (2016) 57, 764–775; Plant J (1998) 16, 681–687). Afterward, we specified NaCl treatment and mannitol treatment for salt and drought stress, respectively, when presenting data in the main text. In some sentences including the conclusions, we mentioned salt and drought stress for NaCl treatment and mannitol treatment, respectively, when implementing our data to salt and drought stress conditions. Please refer to the text for the revision details (there are too many revisions to list here).
Previous comment #7: “No statistical results of RNA sequencing are indicated in the main document.”
Authors response: “We added statistical results, FDR, in Table S2, Table S3, and Table S4.”
My current answer: This makes no sense. 1. What do the authors mean by left reads and right reads?? They should have mapped a consensus…
- Many figures and tables in these new SI files are incomplete and lack information on how they were obtained.
- In addition: the authors are saying that all data are in SI files when obviously raw data should be deposited in a public repository.
(response) We apologize for not being clear in our previous response. We think that we need to clarify our previous response. Because a reviewer had requested to correct all the p-values with metrics like FDR or BH correction, we corrected all the p-values with FDR and updated all the results and conclusions accordingly. After then, we added the statistical significance of mRNA-Seq results to a column (FDR) of Table S2, Table S3, and Table S4. In addition, typically, heatmap results, which visually show the expressions visually, do not include statistical significance (Plant Physiol (2017) 175, 1158-1174; J Exp Bot (2017) 68, 2991-3005; Plant Physiol (2019) 180, 634-653).
- What do the authors mean by left reads and right reads?? They should have mapped a consensus…
(response) We revised Table S1 based on other papers (Plant Cell Physiol (2020) 61, 393-402; Proc Natl Acad Sci USA (2012) 109, 1347-1352; Sci Rep (2016) 6, 37137; Commun Biol (2023) 6, 473), and added information to explain each term in the footnote.
(revision details)
- left reads, right reads -> Forward reads, Reverse reads
- Input, trimmed reads; mapped, mapped reads to reference genome (Arabidopsis genome sequence TAIR 10); aligned pairs, aligned pairs of forward and reverse reads
- Many figures and tables in these new SI files are incomplete and lack information on how they were obtained.
(response) We added more information to figures and tables in SI files. If there is more information to be added, we would appreciate it if the reviewer specifies them.
(revision details)
Figure S1. Expression analysis of osmotic stress-responsive marker genes. Quantitative RT-PCR analysis of RD29A and RAB18 in whole seedlings (a–d) and in roots (e–h) under 150 mM NaCl or 300 mM mannitol treatment for 0, 1, 2, and 4 h. GAPc was used as an internal control. Transcript levels at 0 h were set as 1. Three biological replicates were performed with two technical replicate for each biological replicate. Three independent reactions were performed for each technical replicate. Error bars represent standard deviation (n = 12 reactions). Different letters indicate significant differences (p < 0.05).
Figure S2. Volcano plot of differentially expressed genes (DEGs) under osmotic stress conditions. (a) DEGs in roots under NaCl treatment. (b) DEGs in roots under mannitol treatment. (c) DEGs in whole seedlings under NaCl treatment. (d) DEGs in whole seedlings under mannitol treatment. Red dots and green dots represent upregulated genes and downregulated genes, respectively. Grey dots indicate genes that are not differentially expressed. DEGs were selected by FDR < 0.05 and log2 ratio ≥ 1 condition.
Figure S3. Enriched gene ontology terms of upregulated genes in roots and in whole seedlings under salt and drought stress conditions. (a–c) Enriched Biological Process (BP) (a), Molecular Function (MF) (b), and Cellular Component (CC) (c) of upregulated genes in roots under NaCl treatment. (d–f) Enriched BP (d), MF (e), and CC (f) of upregulated genes in roots under mannitol treatment. (g–i) Enriched BP (g), MF (h), and CC (i) of upregulated genes in whole seedlings under NaCl treatment. (j–l) Enriched BP (j), MF (k), and CC (l) of upregulated genes in whole seedlings under mannitol treatment.
Figure S4. Venn diagrams of DEGs in whole seedlings and in roots. (a) Venn diagram of upregulated genes in whole seedlings and in roots under NaCl treatment. (b) Venn diagram of upregulated genes in whole seedlings and in roots under mannitol treatment. (c) Venn diagram of downregulated genes in whole seedlings and in roots under NaCl treatment. (d) Venn diagram of downregulated genes in whole seedlings and in roots under mannitol treatment.
Figure S6. Expression of transcription factor genes upregulated under osmotic stress conditions in roots. Expression analysis of transcription factor genes upregulated in roots were performed using MeV.
Figure S7. Expression analysis of salt stress-responsive transcription factor gene. Quantitative RT-PCR analysis of RAB18 (a), AT1G62975 (b), AITR5 (c), WRKY29 (d), BOA (e), and MYB3 (f) in whole seedlings, shoots, and roots under 150 mM NaCl treatment for 0, 1, 2, 4, and 8 h. GAPc was used as an internal control. Transcript levels at 0 h in whole seedlings, shoots, or roots were set as 1. Three biological replicates were performed with two technical replicate for each biological replicate. Three independent reactions were performed for each technical replicate. Error bars represent standard deviation (n = 12 reactions). Different letters indicate significant differences (p < 0.05).
Figure S8. Expression analysis of drought stress-responsive transcription factor gene. Quantitative RT-PCR analysis of RAB18 (a), AT1G62975 (b), AITR5 (c), WRKY29 (d), BOA (e), and MYB3 (f) in whole seedlings, shoots, and roots under 300 mM mannitol treatment for 0, 1, 2, 4, and 8 h. GAPc was used as an internal control. Transcript levels at 0 h in whole seedlings, shoots, or roots were set as 1. Two biological replicates were performed with two technical replicate for each biological replicate. Three independent reactions were performed for each technical replicate. Error bars represent standard deviation (n = 6 reactions). Different letters indicate significant differences (p < 0.05).
Table S2. Expression of the ABA-dependent and -independent genes under osmotic stress conditions in whole seedlings and roots
Table S3. Expression of transcription factor genes upregulated under osmotic stress conditions in roots
Table S4. Expression level of alternative splice variants upregulated under osmotic stress conditions in roots
- In addition: the authors are saying that all data are in SI files when obviously raw data should be deposited in a public repository.
(response) We revised Data Availability Statement.
(revision details)
Data Availability Statement: The data presented in this study are available in the Supplementary Materials. The data presented in this study are openly available in Gene Expression Omnibus (GEO); GEO accession number is GSE229217
Previous comment #8: “Several figures have the names in the axis incompleted.”
Authors response: “We revised axis in Figure 1 to show the names completely.”
My current answer: Thank you. But the same still holds for SI files.
(response) We revised the axes of Fig. 3 (g), (i), (l).
Previous comment #9: “How exactly were the ABA-dependent and -independent osmotic stress-responsive genes selected? I found this puzzling since some genes act in ABA- related and also not-related pathways..”
Authors response: “We reviewed published papers (Plant Cell (2016) 28, 2178–2196; Plant Cell (2001) 13, 2063–2083; Plant Cell Physiol (2015) 56, 930–942; Plant Cell (1998) 10, 1391–1406; Plant J (2010) 61, 1041–1052; Curr Opin Plant Biol (2014) 21, 133–139; Plant Cell (2008) 20, 1879–1898; Plant J (2007) 49, 184–193; Plant Mol Biol (2006) 61, 95–109) to identify ABA-dependent and –independent osmotic stress-responsive genes.”
My current answer: My concern still holds as the reader has no idea how the authors choose those genes (not stated anywhere).
(response) We cited the references ABA-dependent and –independent osmotic stress-responsive genes in text to help the readers’ understanding.
(revision details)
Next, we analyzed the expression of well-known ABA-dependent and/or ABA-independent osmotic stress-responsive genes using mRNA-Seq [37–45].
Previous comment #10: “In relation to rt-PCR, the authors indicated "Two technical replicates were performed for each biological replicate. At least two biological replicates showed similar results, with one shown here. Error bars represent standard deviation (n = 6 reactions)." This is simply wrong; the authors cannot choose the replicate they want. That is the reason for having biological and technical replicates....”
Authors response: “We reanalyzed our RT-qPCR results, those shown in Figures 4 and 7, by incorporating all of our biological and technical replicates.”
My current answer: See previous comment concerning biological and technical replicates.
(response) We added the number of technical replicates and biological replicates in each figure legend (Figures 4, 5, 7, and 8). We recently got Figure 5 and Figure 8 with two biological replicates since a review requested the experiments. Data in Figures 5 and 8 support mRNA-Seq data (Tables S3 and S4, Figure S6). Actually, data in Figures 5/8 and mRNA-Seq data support each other. Furthermore, we added the reason/explanation for setting isoform 1 as a control in each Figure legend (Figures 7 and 8).
(revision details)
Figure 4: Three biological replicates were performed with two technical replicates for each biological replicate. Three independent reactions were performed for each technical replicate.
Figure 5: Two biological replicates were performed with two technical replicates for each biological replicate. Three independent reactions were performed for each technical replicate.
Figure 7: Transcript levels at 0 h in isoform 1, the representative model (TAIR, https://www.arabidopsis.org/), were set as 1. Three biological replicates were performed with two technical replicates for each biological replicate. Three independent reactions were performed for each technical replicate.
Figure 8: Transcript levels at 0 h in isoform 1, the representative model (TAIR, https://www.arabidopsis.org/), were set as 1. Two biological replicates were performed with two technical replicates for each biological replicate. Three independent reactions were performed for each technical replicate.
Previous comment #11: “The discussion needs to be updated. There are many genomic studies showing the impacts of osmotic stress in plants, including also Arabidopsis.”
Authors response: “We discussed more about our findings compared to previous meta-analysis studies.”
My current answer: I still have the same concern. The authors have literally added 2 new sentences and as a result the discussion is still very poor. For instance, in relation to the authors aims, which has to find new genes and pathways acting in response to stress in roots of Arabidopsis what are the advances made by this study? How do this connect with other studies?
(response) Thank you for your valuable feedback. In the current revised manuscript, we have additionally discussed potential pathways that they may be involved (as described in lines 400–409). Although we want to understand more functions of newly identified genes and signaling pathways, there are limitations in the current approach. Functional analysis using overexpressing transgenic plants and/or knock-out mutants is required to understand them. We aim to conduct a functional analysis of the newly identified genes to uncover osmotic stress responses that are specific to roots in the future. Additionally, we compared our study to previous meta-analysis studies using microarray to understand possible signaling pathways, in which newly identified genes can be involved in. Transcription factor genes in previous studies are involved in osmotic stress responses in roots through various ways, such as cell wall modification, osmoprotectant synthesis and transport, ROS scavenging, protein metabolism, hormone signaling, and so on, consistent with our study.
Our manuscript is the first to report on genome-wide analysis data of osmotic stress response in Arabidopsis roots using mRNA-Seq. We also found that osmotic stress responses in roots might be mediated by uncharacterized pathways or mechanisms other than well-known ABA-dependent and/or ABA-independent osmotic stress-responsive genes. Furthermore, we identified 26 genes that showed varying expression of alternative splice variants or altered splicing events under salt and/or drought/osmotic stress conditions in Arabidopsis roots
We described these new findings in the Discussion.
Previous comment #12: “Proofreading comments: The authors should carefully revise the text (including the abstract and the title). Many sentences have typos and grammatical mistakes, as well as repetitive words. That occurs throughout the text.”
Authors response: “We had corrected grammatical and/or punctuation issues in the original manuscript (ijms-2360949) and its resubmitted manuscript (current manuscript; ijms-2445482) through English editing service (Editage).”
My current answer: my concern still holds. I’m sorry but this is simply not true. Take a look into the new aims written, as an example: “Given that there are distinct osmotic stress responses in shoots and roots, it suggests that there are the genes responsive to osmotic stresses expressed specifically in the roots.”… This type of situation occurs in many other sentences in the manuscript.
(response) English in the manuscript has been corrected one more time through an English editing service team (Editage: www.editage.co.kr).
I am sorry for not being more positive but there are too many things here that have no support and are not clear to any reader. There are also many details missing that prevent the replication of this study. As a result, I cannot recommend this version for publication, either here or in any other journal.
(response)
We described some methods in more detail to enhance the reader’s understanding as follow:
Line 480: The reaction conditions consisted of 10 min of initial denaturation at 95 °C, followed by repeated cycles at 95 °C for 30 s, 60 °C for 30 s, and 72 °C for 30 s.
Line 525: Gene clustering was performed using MeV ver. 4.9.0 (https://sourceforge.net/projects/mev-tm4/) [67]. Protein network analysis was performed using STRING Apps in Cytoscape ver. 3.9.1 (https://apps.cytoscape.org/apps/stringapp) [68].
